# Thin Arctic sea ice in L-band observations and an ocean reanalysis

Steffen Tietsche[1], Magdalena Alonso-Balmaseda[1], Patricia Rosnay[1], Hao Zuo[1], Xiangshan Tian-Kunze[2], and Lars Kaleschke[2]

[1]European Centre for Medium-Range Weather Forecasts
[2]Institute of Oceanography, University of Hamburg

*Correspondence to:* S. Tietsche (s.tietsche@ecmwf.int)

**Abstract.** L-band radiance measurements of the Earth's surface such as these from the SMOS satellite can be used to distinguish thin from thick ice under cold surface conditions. However, uncertainties can be large due to assumptions in the forward model that converts brightness temperatures into ice thickness, and due to uncertainties in auxiliary fields which need to be independently modelled or observed. It is therefore advisable to perform a critical assessment with independent observational and model data, before using sea-ice thickness products from L-band radiometry for model validation or data assimilation. Here, we discuss version 3.1 of the University of Hamburg SMOS sea-ice thickness data set (SMOS-SIT) from autumn 2011 to autumn 2017, and compare it to the global ocean reanalysis ORAS5, which does not assimilate the SMOS-SIT data. ORAS5 currently provides the ocean and sea-ice initial conditions for all coupled weather, monthly and seasonal forecasts issued by ECMWF. It is concluded that SMOS-SIT provides valuable and unique information on thin sea ice during winter, and can under certain conditions be used to expose deficiencies in the reanalysis. Overall, there is a promising match between sea-ice thicknesses from ORAS5 and SMOS-SIT early in the freezing season (October–December), while later in winter, sea ice is consistently modelled thicker than observed. This is mostly attributable to refrozen polynyas and fracture zones, which are poorly represented in ORAS5 but well detected by SMOS-SIT. However, there are other regions like the Baffin Bay, where biases in the observational data seem to be substantial, as comparison to independent observational data suggests. Despite considerable uncertainties and discrepancies between thin sea ice in SMOS-SIT and ORAS5 at local scales, interannual variability and trends of its large-scale distribution are in good agreement. This gives some confidence in our current capability to monitor climate variability and change of thin sea ice. With further improvements in retrieval methods, forecast models and data assimilation methods, the huge potential of L-band radiometry to derive the thickness of thin sea ice in winter will be realized and provide an important building block for improved predictions in polar regions.

## 1 Introduction

Sea ice has been regularly observed by satellites since the late 1970s. The observations most widely used in the context of large-scale weather and climate models are passive microwave radiance in the range between 6 and 90 GHz. These observations have continuous daily pan-Arctic coverage at a resolution of 50 km or better. However, because of the very small penetration depth of microwave radiation into sea ice at these frequencies, these observations only provide information about the fraction of an area covered by sea ice, not about its thickness.

Considering the importance of sea-ice thickness for atmosphere–ocean surface heat fluxes, and for predicting the further evolution of the sea-ice cover, information about it is indispensable. Substantial heat conduction occurs through thin sea ice in winter, when the temperature contrast is large between the cold surface atmosphere and the relatively warm ocean water below the ice. Approximate calculations show that surface heat fluxes resulting from heat conduction through thin sea ice can easily
reach $100\,\mathrm{Wm^{-2}}$. Predicting the evolution of the sea-ice cover days to months ahead also crucially depends on the sea-ice thickness: thin ice will evolve much more quickly than thick ice because it is more susceptible to dispersion or compression by winds, and because the larger surface heat fluxes it allows can change the mass of ice much faster.

The thickness of sea ice is much harder to derive from satellite observations than its area coverage, and each of the existing methods has its own strong limitations. Infrared emission measurements of the ice surface temperature (Wang et al., 2010; Yu
and Rothrock, 1996; Mäkynen et al., 2013) only work for very thin ice without snow cover, and can only be used under cloud-free conditions. Laser and radar altimetry (Kwok and Cunningham, 2008; Laxon et al., 2013; Ricker et al., 2014) suffer from high measurement noise and narrow foot-prints, and become unfeasible for thicknesses below 0.5 m and in the presence of surface waves. Finally, the thickness of thin sea ice can be derived from L-band microwave radiance measurements (Kaleschke et al., 2012; Tian-Kunze et al., 2014; Mecklenburg et al., 2016). This method allows daily pan-Arctic coverage for ice thickness
of up to 1 m with about 30 km spatial resolution. It requires, however, a complex radiative transfer model, which means that calculated emissivities can be sensitive to the retrieval assumptions and auxiliary fields used.

Radiance measurements of L-Band brightness temperatures (TB) from space have for the first time become available with ESA's SMOS mission launched in 2009. There is a high sensitivity of L-Band TB to the thickness of thin sea ice, but a reliable retrieval of sea-ice thickness depends on high-quality constraints on the other parameters which the TB are sensitive to –
most importantly, sea-ice concentration, and temperature and salinity profiles within the ice (Tian-Kunze et al., 2014). These external data dependencies introduce uncertainties that are often difficult to quantify. For instance, near-surface temperature over Arctic sea ice can vary by several degrees between atmospheric analyses from different centres (Bauer et al., 2016). Moreover, different radiative transfer models exist to calculate the L-band emissivity of a given sea ice slab, and the calculated L-band TB can vary considerably depending on the model chosen (Maaß, 2013; Richter et al., 2018).

For prognostic sea-ice models as included in climate and numerical weather forecasting models, simulating thin sea ice is challenging as well. Although climate models have been including prognostic sea ice for many years, two factors limit their usefulness for investigating thin sea ice. First, sea-ice thickness is often represented in a mono-category approach similar to that in Fichefet and Maqueda (1997), with very simplified treatment of thin sea ice (although in the latest generation of climate models there is a clear trend towards a multi-category approach to simulate ice thickness (Notz et al., 2016)). Second, thin-
sea-ice features are often short-lived (a few days or less) and local in scale (smaller than 100 km). These temporal and spatial scales are usually not well resolved in climate models, whose output tends to be monthly-mean fields on grids with cell sizes of 100 km or more.

Prognostic sea-ice models as included in numerical weather forecasting models are usually run at higher spatial resolution (e.g. around 10-15 km in the Arctic for the setup discussed in this study), and usually their output is analysed based on daily-
mean or instantaneous values. Thus, they clearly resolve many of the small-scale, short-lived thin sea ice features. However,

they often use the same simplified mono-category approach towards simulating ice thickness, and hence suffer from the same structural problems as the sea-ice component in climate models.

These prognostic models are combined with observations using data assimilation, to arrive at the best estimate of the true state, the so-called analysis. If the same system is applied to observations spanning multiple years, it is usually called a re-analysis, a convention which we will follow here. State-of-the-art ocean reanalyses employ prognostic sea-ice models at relatively high spatial resolution as suitable for numerical weather prediction. These ocean reanalyses have many users (see e.g. (Le Traon and Others, 2017)), who might not all have the resources to carry out an assessment how the reanalysis product compares to observations. Such overall assessments of several reanalyses have been carried out in the past (Balmaseda et al., 2015; Chevallier et al., 2016; Uotila et al., 2018), but have not addressed the specific issue of thin sea ice.

This study aims to provide an overview assessment of agreements and discrepancies of sea-ice thickness between an observational product from L-band radiometry on the one hand, and a ocean reanalysis that does not assimilate these observations on the other hand. This assessment is a first necessary step towards the eventual assimilation of these observational data, because large systematic errors in either the observations or the forecast model will make successful data assimilation difficult. Previous studies report overall slightly positive results when assimilating L-band sea-ice thickness observations (Yang et al., 2014; Xie et al., 2016), but without doubting the validity of the observational data. As we will show here, both reanalysis and observations can contain large and systematic errors. We argue that these need to be characterized, understood, and properly treated in any future data assimilation system in order to obtain an improved estimate of the true sea-ice thickness.

Being an overview assessment, this study provides guidance and inspiration for future research by identifying the characteristic main agreements and discrepancies between sea-ice thickness from L-band retrievals and an ocean reanalysis. We offer plausible hypotheses for the identified discrepancies and are able to verify some of them quantitatively. However, due to the nature of our methods, there are many discrepancies where we cannot offer conclusive evidence of their root causes. This would require systematic numerical experimentation with the retrieval and reanalysis models, a substantial technical, computational and analytical effort that is beyond the scope of the diagnostic overview study presented here. First steps in this direction have already been taken by Maaß (2013) and Richter et al. (2018), who perform sensitivity experiments with the retrieval model, and by Zuo et al. (2015) and Shi and Lohmann (2017), who perform sensitivity experiments with the forecast model and data assimilation methods.

The remainder of the paper is structured as follows: we start with a description of the methods used to produce the observational sea-ice thickness product SMOS-SIT in Section 2.1, and the ocean reanalysis system ORAS5 in Section 2.2. The pan-Arctic reanalysis–observation departures are discussed in Section 3, followed by a more detailed discussion of the regional differences in Section 4. Section 5 makes the point that, despite often large reanalysis–observation departures, climate variability and trend of the thin sea-ice area are in broad agreement between reanalysis and observations. A discussion of the results is presented in Section 6, and Section 7 summarizes the main results. More detailed technical information and discussion of the limits of SMOS-SIT can be found in Appendices A–D.

## 2 Model and data

### 2.1 SMOS-SIT sea ice thickness product

Thin sea ice thickness (nominal cut-off at 1.5 m) has been retrieved at the University of Hamburg from L-band brightness temperatures (TB) at 1.4 GHz measured by the MIRAS radiometer on board of SMOS. The retrieval algorithm consists of a

thermodynamic sea ice model and a one-ice-layer radiative transfer model (Kaleschke et al., 2012; Tian-Kunze et al., 2014). The resulting plane layer thickness is multiplied by a correction factor assuming a log-normal thickness distribution (Tian-Kunze et al., 2014). The algorithm has been used for the production of a SMOS-based sea ice thickness data set in polar-stereographic projection in 12.5 km grid resolution from 2010 on (http://icdc.cen.uni-hamburg.de/1/daten/cryosphere/l3c-smos-sit.html) (Tian-Kunze et al., 2014). In this study, we use the most up-to-date version (v3.1, based on v620 L1C brightness tempera-

tures), which has been produced operationally since October 2016. The v3.1 data for the previous winter seasons had been reprocessed using the same algorithm. In the beginning two years of SMOS operation, the signals were strongly influenced by Radio Frequency Interference (RFI), so we exclude the winter 2010/2011 from our discussion. Previous versions of the algorithm have been described in Kaleschke et al. (2012), Tian-Kunze et al. (2014), and Kaleschke et al. (2016), who also provide comparison to EM-bird measurements, infrared-derived, and modelled sea ice thickness.

Brightness temperature used in the algorithm is the daily mean intensity, which is the average of horizontal and vertical polarization. Over sea ice, the intensity is almost independent of incidence angle. The average over the incidence angles 0-40° is taken, in order to reduce the brightness temperature uncertainty to about 0.5 K. In the algorithms prior to v3.1, RFI contaminated snapshots have been discarded using a threshold value of 300 K, applied either to horizontal or vertical polarization. However, in v3.1 the new quality flags given in the v620 L1C data have been implemented to identify the data contaminated not only by

RFI but also by sun, or by geometric effects.

The retrieval method needs additional auxiliary data as boundary conditions for the thermodynamic as well as the radiation model: bulk ice temperature is estimated from surface air temperature extracted from the JRA-55 atmospheric reanalysis (Ebita et al., 2011). Bulk sea-ice salinity is calculated with the methods described in Tian-Kunze et al. (2014) based on a weekly climatology of sea surface salinity from a simulation with the MIT General Circulation Model (Marshall et al., 1997) covering

the years 2002–2009. Brightness temperatures over sea ice depend on the dielectric properties of the ice layer, which vary with ice temperature and ice salinity (Menashi et al., 1993; Kaleschke et al., 2010, 2012). The temperature profile within the ice is assumed to be linear, which is a good approximation for thin ice and slow changes in the meteorological conditions. The retrieval algorithm works only under cold conditions: the presence of surface melting invalidates the retrieval assumptions.

Ice thickness retrieval uncertainties are given pixel-wise each day in the data set. There are several factors that cause un-

certainties in the sea ice thickness retrieval: the uncertainty of the SMOS TB, the uncertainties of the auxiliary data sets, the uncertainties in ice temperature and ice salinity, and the assumptions made for the radiation and thermodynamic models, for example 100% ice coverage.

The uncertainty of daily mean TB is mostly less than 0.5 K, except for the years 2010 and 2011, when, due to RFI problems, the percentage of RFI contaminated TB measurements was relatively high near the coasts of Russia and Greenland. The

uncertainties caused by bulk ice temperature and bulk ice salinity depend on the uncertainties of surface air temperature and sea surface salinity, which are the boundary conditions in the retrieval. As a first approximation, a sea-ice surface temperature uncertainty of 1 K has been assumed. The uncertainty of sea surface salinity is estimated from standard deviation of an ocean simulation for the years 2002-2009 (Tian-Kunze et al., 2014).

In addition to the uncertainty factors discussed above, version 3.1 of SMOS-SIT also considers the uncertainty in the fitted parameter $\sigma$ of the assumed log-normal distribution for the subgrid-scale sea-ice thickness (Kaleschke et al., 2017). The fit uncertainty is the standard deviation of the natural logarithm $\ln \sigma$, and it is derived from six years of NASA OIB airborne observations of ice thickness (Kurtz et al., 2013). The average ice thickness uncertainty from this contribution is less than 0.1 m.

The total ice thickness uncertainty provided in SMOS-SIT is the sum of the above-mentioned uncertainties of TB, ice temperature and salinity, and ice thickness distribution function. Errors caused by assumptions on heat fluxes and snow thickness have not yet been included. The radiation model used in the retrieval is a one-layer model. Thus, with this radiative transfer model, it is not possible to discuss the impact of ice temperature and salinity profiles on the ice thickness retrieval. Generally, the uncertainty increases with increasing ice thickness. For thinner ice the relationship between ice thickness and ice thickness

uncertainty is almost linear. A fit function between ice thickness and ice thickness uncertainty is derived from one winter period of SMOS data. This function is then implemented in the retrieval for the calculation of ice thickness uncertainty.

    In addition to the retrieval uncertainty, the data set contains the so-called saturation ratio for each SMOS pixel, which gives a useful estimate of the sensitivity of SMOS brightness temperature to ice thickness for the values of the auxiliary fields valid for the SMOS pixel. The saturation ratio is defined as the ratio of the retrieved ice thickness to the maximal retrievable ice

thickness, which is reached when SMOS brightness temperature changes less than 0.1 K per cm ice thickness change (Tian-Kunze et al., 2014).

    For more detailed technical information and a discussion of the limits of SMOS-SIT please refer to the Appendices. Appendix A shows that there are some substantial differences in the SMOS-SIT data set between the current version 3.1 and the previous version 2.3. In Appendix B, the fundamental limits of retrieving sea-ice thickness from SMOS brightness tempera-

tures are touched upon, and evidence for these limits from the data themselves is presented. Appendix C discusses unrealistic day-to-day fluctuations in retrieved sea-ice thickness, and Appendix D demonstrates that using SMOS-SIT without removing high-uncertainty data points can lead to wrong conclusions when studying year-to-year variability of thin sea ice.

## 2.2 ORAS5 sea-ice–ocean reanalysis

The ECMWF ocean reanalysis system 5 (ORAS5) is a state estimate of the global ocean and sea ice from 1979 to today, and is

being used to provide ocean and sea ice initial conditions for operational forecasts at ECMWF (Zuo et al., 2017).

    The NEMO ocean model version 3.4.1 (Madec, 2008) has been used for ORAS5 in a global configuration with a tripolar grid with a resolution of 1/4 degree at the equator. One of the poles of the grid is located on the Antarctic continent, and the other two are in Central Asia and North Canada. Horizontal resolution in northern high latitudes ranges from less than 5 km

(Canadian Archipelago south of Victoria Island) to about 17 km (Bering Sea and Sea of Okhotsk). There are 75 vertical levels, with level spacing increasing from 1 m at the surface to 200 m in the deep ocean.

ORAS5 contains the dynamic-thermodynamic sea ice model LIM2 (Fichefet and Maqueda, 1997). The sea ice model is run with a viscous-plastic rheology. LIM2 has fractional ice cover, a single ice thickness category (Hibler III, 1979), and calculates vertical heat flux within the ice according to the three-layer Semtner scheme (Semtner, 1976). Snow on sea ice is modelled, but melt ponds are not.

The single-thickness approach of LIM2 necessitates a very simplified treatment of open-water sea-ice formation: as in Hibler III (1979), a critical ice thickness $h_0$ is introduced that distinguishes "thin" from "thick" ice. In ORAS5, $h_0$ is equal to $0.6\,\text{m}$ in the Arctic. The critical ice thickness determines how fast the ice concentration increases under freezing conditions, and is therefore also called the lead-close parameter (see Smedsrud and Martin (2015)). In a model grid cell that was previously ice-free, new sea ice forms thermodynamically at a constant actual floe thickness that is equal to $h_0$. This is obviously an overly simplistic representation of how sea ice really forms from open water: by formation and solidification of grease ice (Smedsrud and Martin, 2015). This modelling assumption introduces an artificially increased frequency of occurrence of grid-cell mean ice thickness around $h_0$ under freezing conditions because growth rates for grid-cell mean ice thicknesses below $h_0$ are over-estimated. New generations of sea-ice models, for instance LIM3 (Vancoppenolle et al., 2009) or CICE5 (Hunke et al., 2015) have a much smaller and state-dependent $h_0$, which avoids this problem.

Forcing fields for ORAS5 are derived from the atmospheric reanalysis ERA-Interim (Dee et al., 2011) until the end of 2014, and from the operational ECMWF atmospheric analysis from the beginning of 2015 on. Sea surface temperature for years from 2008 on is constrained to observations from the UK Met Office Operational Sea Surface Temperature and Sea Ice Analysis (OSTIA) by a strong restoring term. Assimilation of subsurface ocean temperature and salinity, of sea ice concentration and sea level anomalies is performed using a 3DVar-FGAT procedure (Daget et al., 2008). The length of the data assimilation window is 5 days.

Sea-ice concentration in ORAS5 is assimilated from the level-4 OSTIA product (Donlon et al., 2012). OSTIA sea-ice concentration is created by interpolating and in-filling the sea-ice concentration product of the EUMETSAT Ocean and Sea Ice Satellite Application Facility (http://osisaf.met.no/p/ice) to a global regular grid with 1/20 degree resolution and filling in missing values. The sea-ice concentration assimilation is univariate with no direct impact on the floe ice thickness. However, grid-cell mean ice thickness is directly affected by the assimilation increments (see Tietsche et al. (2013) for details). There is no assimilation of sea-ice thickness observations in ORAS5, so it is completely independent of SMOS-SIT.

ORAS5 consists of five ensemble members which are obtained by perturbing the surface forcing, and by assimilating perturbed observations (see Zuo et al. (2017) for details).

For a full description of the immediate predecessor of ORAS5, see the documentation of ORAP5 in Zuo et al. (2015); Tietsche et al. (2017). ORAP5 has been found to simulate well the overall ice thickness in the Arctic in comparison with other state-of-the-art ocean reanalyses (Uotila et al., 2018).

## 3 Pan-Arctic reanalysis–observation departures

The SMOS-SIT data provide essential information about sea ice that is complementary to observation of sea ice concentration using higher-frequency passive microwave channels. To illustrate that, Figure 1 shows SMOS-SIT sea-ice thickness together with observed sea-ice concentration from the OSTIA product for a day early in the freezing season, and for a day late in the freezing season. Please not that here and elsewhere, sea-ice thickness denotes the *grid-cell mean* sea-ice thickness for both SMOS-SIT and ORAS5. Early in the freezing season, there are large areas of newly-formed sea ice that is thin. Figure 1(a) shows that sea ice thickness of $0.6 - 0.7$ m dominates in the Beaufort and Chukchi Seas, as well as part of the central Arctic Ocean adjacent to them. In the Baffin Bay, sea ice thickness from SMOS-SIT is even thinner, at around $0.2 - 0.3$ m. All these regions exhibit sea-ice concentrations of virtually 100% (Figure 1b), which demonstrates that sea-ice concentration observational products like OSTIA can not be used to distinguish areas with thin new sea ice from areas of old thick sea ice; sea-ice thickness observational products like SMOS-SIT are needed to do that.

[Figure 1 about here.]

Sea-ice thickness in ORAS5 in early winter is comparable with that of SMOS-SIT (Figure 1c). However, the model tends to simulate thicker ice on average. Note that the departures in Figures 1c,f are only shown for SMOS-SIT data points with a saturation ratio less than 90% and total retrieval uncertainty of less than 1 m (see Section 2.1 for definitions of these). Positive departures dominate, especially close to regions of thick ice. There are a few places in the Beaufort and the Siberian Shelf Seas with negative departures, but in most of the thin-ice areas ORAS5 simulates ice around $0.4$ m thicker than retrieved by SMOS-SIT.

As the freezing season progresses, the ice edge moves further south outside of the Arctic Basin, and previously formed thin ice in the Arctic Basin becomes thicker. Polynyas and fracture zones begin to form. These re-freeze very quickly, which is evident by the near-100% sea-ice concentration but greatly reduced sea-ice thickness in these features. Figure 1(d) shows extensive refrozen polynyas in the Kara and Laptev Seas, as well as a fracture zone covering the whole Beaufort Sea. In the Baffin Bay, sea-ice thickness derived by SMOS-SIT is mostly below 0.3 m. Again, none of these features within the ice pack are picked up by the sea-ice OSTIA concentration product, which shows homogeneously high ice concentration throughout the ice pack (Figure 1e).

The departures between ORAS5 and SMOS-SIT in late winter are large and positive throughout (Figure 1f), with values of 1m or more dominating. Most of this is due to the failure of the reanalysis to simulate relevant features like the refrozen coastal polynya in the Laptev Sea, or refrozen fracture zones like the one visible in the SMOS-SIT data for the Beaufort Sea.

There are multiple plausible reasons for the poor representation of refrozen polynyas and fracture zones in the reanalysis: various deficiencies in the ocean and sea-ice models (e.g. too thick ice, inappropriate rheology, insufficient modelling of open-water ice growth, too strong upper-ocean stratification), the data assimilation methods (e.g. inappropriate background error covariance between ice concentration and ice volume), or deficiencies in the atmospheric forcing (e.g. too weak off-shore winds). Further investigation of this reanalysis deficit is clearly needed, but for the most part this requires dedicated experimentation and is therefore out of the scope of this study. However, it can be said that there are conspicuous features in

maps of sea-ice concentration increments (not shown), which directly affect grid-cell mean ice thickness through implied ice volume increments as discussed by Tietsche et al. (2013). For the day in question, 15 April 2016, the sea-ice concentration increments are large and positive in the refrozen polynyas and fracture zones. This would suggests that the model dynamics tend to produce the features, but the assimilation increments suppress them in the reanalysis.

In the Barents Sea there is good agreement between ORAS5 and SMOS-SIT, with a positive departure of 20 cm or less. Finally, the Baffin Bay stands out as having extensive thin ice cover in SMOS-SIT, but thick ice in ORAS5. The North Water Polynya at the northern end of Baffin Bay is captured both by SMOS-SIT and ORAS5.

The previous example maps show typical conditions in early and late winter, and typical departures between ORAS5 and SMOS-SIT. For a more quantitative assessment, we calculate departures for co-located daily sea-ice thickness in (a) the early-

winter period 15 October to 15 December for the years 2011–2017, and (b) the late-winter period 15 February to 15 April for the years 2012–2017. We exclude data points where the SMOS-SIT retrieval is known to be unreliable: data points with a retrieval uncertainty of more than 1 m, a saturation ration of above 90%, or a sea-ice concentration below 30% are not considered (see Section 2.1 for explanations of retrieval uncertainty and saturation ratio).

From these co-located pairs of observed and modelled daily sea-ice thicknesses we calculate the normalized bivariate joint

frequency distribution, which we will call *scatter density* in the following for the sake of brevity. Scatter density plots give a quite complete picture of the departure statistics. For a good match, density should be high on the one-to-one line and low elsewhere. High density above the one-to-one line indicates positive bias, high density below the one-to-one line indicates negative bias. Conditional departure characteristics e.g. for a certain range of observed values can also easily be derived visually.

20                                    [Figure 2 about here.]

As can be seen from the scatter density in Figure 2a, in early winter the agreement between SMOS-SIT and ORAS5 sea ice thickness is quite promising as the distribution is close to the one-to-one line. However, the overestimation of sea-ice thickness by ORAS5, which was already visually apparent from the maps in Figure 1, is confirmed. For observed sea-ice thickness between 0 and 0.3 m, ORAS5 sea-ice thickness is about 0.3 m higher. The agreement becomes better for higher observed sea-

ice thickness in the range 0.5-1 m. Note that the scatter density distribution has wide tails. For instance, for an ice thickness of 0.4 m in SMOS-SIT, ORAS5 values of up to 1.5 m exist. This is not so obvious in the scatter density, but is clearly visible in the corresponding scatter plot that tends to highlight outlier data points (not shown).

Part of the reason for ORAS5 having higher ice thickness than SMOS-SIT early in the freezing season is the simplified representation of thin ice in LIM2, the sea-ice component of ORAS5 (see Section 2.2): thermodynamic formation of new ice in

LIM2 happens at a fixed actual (floe) thickness of 0.6 m, a value that has been chosen to approximate growth processes in the presence of thick sea ice (Hibler III, 1979). Quite obviously, this is not a good representation of how sea ice forms from open water, which is the dominant regime at the ice edge early in the freezing season. As can be seen in Figure 8c, the simplified LIM2 treatment of thin sea ice leads to an artificially high frequency of occurrence of grid-cell mean ice thickness around

0.6 m, because ice growth rates are artificially enhanced for grid-cell mean ice thicknesses below that value (see Mellor and Kantha (1989), Tietsche et al. (2017) and Shi and Lohmann (2017) for further discussion on this).

A second reason for higher ice thickness in ORAS5 than SMOS-SIT early in the freezing season is that SMOS-SIT retrieves sea-ice thickness under the assumption of 100% sea ice concentration. If the area captured by a SMOS pixel has only partial ice cover, the SMOS-SIT ice thickness retrieval is biased thin (Tian-Kunze et al., 2014). As can be seen from Figure 9b, there is an almost perfect linear relationship between SMOS TB and sea-ice concentration for intermediate sea-ice concentration values, which clearly indicates that geometrical averaging of open-water and sea-ice emissivity within a SMOS pixel is playing a role. When excluding data points from Figure 2a where sea-ice concentration is below 95% (not shown), the scatter density conditional on SMOS-SIT thickness being below 0.2 m is almost zero, which is good indication that all thickness retrievals at least up to this thickness are likely to be biased low due to neglecting the open-water contribution to L-band emissivity.

In late winter, ORAS5 has much higher sea-ice thickness than SMOS-SIT (Figure 2b). Departures between 0.5m and 1m are common throughout the SMOS-SIT thickness range of 0–1m. There is a more linear shape of the scatter density distribution – this is promising in principle, but could result from compensating errors in different regions, which would make the relationship less relevant. The scatter distribution is also much wider than for early winter, indicating larger and more uncertain reanalysis– observation differences. The larger discrepancy in later winter has several causes. Figure 1(d-f) illustrate the most obvious one: the ocean reanalysis does not simulate polynyas and fracture zones well. But there are other causes, some of which are related to the properties of SMOS-SIT data. In the following Section, we analyse the late-winter departures in more detail.

## 4   Regional contrasts

There is considerable regional dependence of the departures in late winter (February to April). Figure 3 shows the SMOS-SIT/ORAS5 scatter density as in Figure 2b), but for three key regions separately: the Barents and Kara Seas, the Laptev Sea, and the Baffin Bay. For the Barents and Kara Seas (Figure 3a), the departure statistics are almost as good as for the pan-Arctic in early winter (Figure 2a). We can conclude that this region has relatively good agreement between ORAS5 and SMOS-SIT sea ice thickness throughout the winter. In the Laptev Sea (Figure 3b), ORAS5 has no ice thickness below 1 m, whereas SMOS-SIT detects a lot of ice thinner than 1 m. There is a very low correlation between ORAS5 and SMOS-SIT ice thickness. This behaviour is consistent with our earlier assessment that refrozen polynyas do occur frequently in the Laptev sea in late winter, and that they are detected by SMOS-SIT but not well represented in ORAS5.

Finally, Figure 3c shows the late-winter scatter density for the Baffin Bay, which again has characteristics that are very different from the other two regions. In general, ORAS5 simulates much thicker ice than retrieved by SMOS-SIT, but in contrast to the Laptev-Sea case, there is a quite high rank correlation between SMOS-SIT and ORAS5, i.e. higher SMOS-SIT values are often associated with higher ORAS5 values but the correspondence is not necessary linear. This suggests systematic rather than random sources for the departures.

[Figure 3 about here.]

An interpretation of the results in Figure 3 needs to start from the appreciation that the regions shown have quite different physical characteristics: in the Barents and Kara Seas, sea ice is strongly affected by warm Atlantic water being advected towards and under the ice, which means the ice cover is constrained by SST. At the same time, prevailing winds modulate the location of the ice edge by transporting the ice. Both processes are expected to be reasonably well simulated by ORAS5, because winds are prescribed as forcing, and the SST are ingested from an observational product. From the observational side, most of the calibration and validation campaigns for SMOS-SIT have been carried out in this area (Kaleschke et al., 2016). Thus, the Barents and Kara Seas can be expected to be the region where the reanalysis-observation agreement is best.

In the Laptev Sea, sea ice is still relatively well observed when it comes to SMOS-SIT validation, but it is more difficult to simulate in ORAS5. Because there is no ice edge in the Laptev Sea, SST information cannot be used to constrain the ice cover. Furthermore, as clearly visible in Figure 1, extensive polynyas form there in February to April, mainly when offshore winds push back the ice from land or land-fast sea ice. These processes are not well simulated by the reanalysis, which tends to keep a compact thick sea ice cover even in the presence of offshore winds. As a result, major departures can be expected.

In the Baffin Bay, the occurrence of thinner ice of varying thickness is modelled and observed, but the modelled ice is roughly twice as thick. There is independent information that suggests that SMOS ice thickness is biased low there (see Laxon et al. (2013); Landy et al. (2017); Tilling et al. (2015)). CryoSat2 estimates (http://www.cpom.ucl.ac.uk/csopr/seaice.html) indicate that between February and April, the ice in this region is typically 1.5 m thick. This is confirmed by independent expert judgement by ice analysts, who estimate that ice in this region and this season would typically be at least 1m thick (Nick Hughes, personal communication).

To further illustrate and consolidate the findings from Figure 3, we plot time series for two representative locations in the Laptev Sea and the Baffin Bay in Figure 4. Both show the typical behaviour of reanalysis–observation departures: SMOS-SIT and ORAS5 match well early in winter, but later on ORAS5 ice keeps getting thicker while SMOS-SIT thickness saturates, albeit with some strong fluctuations. We choose to present a full freezing season in the winter 2011/2012, because this allows co-location with independent data in both locations. For the Laptev Sea (Figure 4a), there was a campaign in April that measured the ice thickness using a so-called EM-bird. This measurement method has demonstrated uncertainties of less than 0.1 m (Haas et al., 2009), hence we can use it as the "ground truth" to benchmark remote sensing observations and reanalysis results. The EM-bird measurement confirms that the ice was indeed only about 0.5 m thick, which indicates the presence of new thin ice in the Laptev-Sea Polynya (Tian-Kunze et al., 2014). The reanalysis is not able to simulate that. The CryoSat2 estimate for this location is around 1m averaged over March and April, halfway between ORAS5 and SMOS-SIT.

For a representative location in the Baffin Bay (Figure 4b), there is reasonable match between reanalysis and observations until January. After that, the sea ice in ORAS5 keeps growing to reach thicknesses of 1.5 – 2 m in mid-April, whereas SMOS-SIT observations level off between 0.5 and 1m until mid-April. The CryoSat2 estimate for this location and averaged over March/April 2012 is 1.8m. This behaviour is generic: it occurs in all years, and also when considering an area average over the Western Baffin Bay as defined by Landy et al. (2017)

When judging compatibility of observational and model-based estimates of sea ice thickness, their uncertainties should be taken into account. The available uncertainty estimates are indicated in Figure 4 in the form of five perturbed ensemble members

of the ORAS5 reanalysis, and in the form of lower and upper bounds of the SMOS-SIT uncertainty estimate provided with the data set. The estimated ORAS5 uncertainty is very small – well below 0.3 m most of the time. It is almost certainly too small, as it does not account for structural uncertainty in the model and data assimilation methods. By contrast, the SMOS-SIT uncertainty range (see Section 2.1) is very variable, and often very large. Sometimes it covers the whole range of fathomable values; sometimes it is small, but independent evidence suggests that the truth lies far outside the uncertainty range provided. An example of the former case is the SMOS-SIT ice thickness in the Laptev Sea (Figure 4a) in February: the retrieved value is 1.2 m, but the uncertainty range goes from 0m to more than 2m. An example for the latter case is the SMOS-SIT ice thickness in the Baffin Bay in April 2012 (Figure 4b): the retrieved value is 0.5 m with an uncertainty estimate of only 0.1 m. As argued before, the true sea ice thickness was very likely much higher than that.

[Figure 4 about here.]

Given that ORAS5, CryoSat2, and expert judgement agree that sea ice in the Baffin Bay in this time of the year should be considerably thicker than SMOS-derived thicknesses, we tentatively suggest that there is a problem with the retrieval assumption of SMOS-SIT in this region. From Figures 5 (a),(e) it can be seen that the slight decrease in SMOS TB from February onwards is interpreted as a strong decrease in retrieved sea-ice thickness in SMOS-SIT, in disagreement with the ORAS5 reanalysis and independent observations.

A sea ice concentration slightly below 100% (Figure 5b) might also play a role: For the high SMOS TB typical of late-winter conditions in the Baffin Bay, even a few percent of open water within the SMOS footprint will lower TB significantly by geometrical averaging (note that differences of open-water fraction of a few percent are difficult if not impossible to observe reliably (Ivanova et al., 2015)). The assumption of 100% sea-ice cover made by SMOS-SIT will then lead to a thickness retrieval that is biased low. The scatter density of OSTIA sea-ice concentration versus SMOS-SIT sea-ice thickness for the Western Baffin Bay in late winter (not shown) shows moderate correlation between the two, i.e. there was open water present and it was usually associated with lower ice thicknesses in the SMOS-SIT retrievals. This is an indication – but not proof – that SMOS-SIT might systematically underestimate ice thickness in the Baffin Bay because of non-negligible amounts of open water.

Sea-ice surface temperature (Figure 5d) is almost always colder in ORAS5 than in SMOS-SIT. This is consistent with the ice being thicker in ORAS5 than in SMOS-SIT: the thicker the ice, the smaller the surface heating by conductive heat flux from the relatively warm ocean water below the ice to the relatively cold surface of the ice. However, different near-surface temperatures in the two reanalyses used (JRA-55 and ERA-Interim,) might also play a role (see Bauer et al. (2016)), because they will have a direct impact on the implied sea-ice bulk temperature. Note that there is an apparent artefact in the ice surface temperature in the SMOS-SIT product: it has a constant value of around -4°C for extended periods in November and December. Differences in snow thicknesses (Figure 5c) mirror differences in the ice thickness, because SMOS-SIT assumes an empirical piecewise linear relationship between the two (Tian-Kunze et al., 2014). Furthermore, sensitivity studies by Maaß (2013) suggest that the decrease in TB could be the result of the sea ice becoming fresher at a different rate than assumed by the empirical rate assumed by SMOS-SIT. Testing all these hypothesis in detail is beyond the scope of this paper, because neither does SMOS-SIT deliver

the assumed sea ice salinity as part of the data product, nor does the ORAS5 sea ice model have a good treatment of ice salinity. Further investigation should be undertaken, and we suggest that the assumed sea ice salinity be made part of the SMOS-SIT data product.

[Figure 5 about here.]

## 5 Interannual variability

Despite the uncertainties at a local scale discussed in the previous sections, there is good agreement in the large-scale distribution of thin sea ice and its interannual variability. Figure 6 shows time series of the area covered by sea ice with thickness above various thresholds in November from 2011 to 2017. The uppermost curve is the area of sea ice with at least 0.1 m thickness. The 0.1 m curve corresponds quite well to the NSIDC sea ice extent if the observational gap around the North Pole is taken into account. The lowermost curve is the area of sea ice with at least 0.9 m thickness.

The overall magnitude, variability and trend of the area for the various ice thickness thresholds generally agree quite well between ORAS5 and SMOS-SIT. The extreme summer minimum in 2012 is visible as reduced sea ice area in November for all thickness classes. In 2013, there was a marked recovery. Since then, there has been a downward trend in all classes, with a small uptick in November 2017. Importantly, this indicates that the well-established summer sea ice decline in recent years has started to affect the winter-time state. These signals of interannual variability are in good agreement with ice volume estimates derived from CryoSat2 radar altimetry (Tilling et al., 2015).

It is important to recall that, in the thickness range 0.9 m and above, SMOS-SIT relies heavily on auxiliary fields to retrieve the sea-ice thickness from SMOS brightness temperature. To produce Figure 6 it was necessary to consider all SMOS-SIT data points, even those with high uncertainty and/or saturation ratio close to 100%. As shown in Appendix D, the resulting maps and scatter densities are not realistic, and one should be cautious when interpreting the lowermost curve in Figure 6a. Nevertheless, it is encouraging to see that overall the same interannual variability and trends of thin sea ice area are derived from ORAS5 and SMOS-SIT.

[Figure 6 about here.]

Interannual variability and trends for sea ice in the Arctic do not occur in synchrony in different regions. Figure 6 shows November conditions, when sea ice is present not only in the central Arctic Ocean, but also in the adjacent Seas, in the Canadian Archipelago, The Baffin Bay, Labrador Sea and the Hudson Bay. All these regions are exposed to regional climate variability and change that is not necessarily aligned: the Barents, Kara and Laptev Seas are heavily influenced by the North Atlantic inflow. In the East Siberian, Chukchi and Beaufort Seas the role of the North Atlantic diminishes, and other processes related to the Siberian High and Pacific climate are more important.

In the East Siberian, Chukchi, and Beaufort Seas (Figure 7a,b), interannual variability of area cover is higher for thicker ice than it is for thinner ice. This feature is detected by both SMOS-SIT and ORAS5; it is more pronounced in ORAS5, where the

area covered by ice thicker than 0.7 m more than doubled between 2012 and 2013, and then decreased in each subsequent to reach the same level as 2012 in 2016.

The Barents, Kara and Laptev Seas (Figure 7c,d), also exhibit a strongly reduced area coverage in 2012 for all thickness categories. However, ice cover continued to increase until 2014, by which time the area covered was almost twice as high as 2012 in some categories. The unusually high area cover in 2014 might at least in parts be due to an unusual circulation in autumn 2014: anomalously high pressure over Scandinavia combined with low pressure over Siberia in September-November led to anomalous high northerly components in the winds in these seas, which would have both encouraged thermodynamic ice growth and spreading of the ice by advection.

Another interesting feature in the Barents, Kara, and Laptev Seas is the increasing area of ice thicker than 0.9 m simulated by ORAS5. The year-to-year changes in thicker ice area as seen by SMOS-SIT are very different, but we would advise caution when interpreting the SMOS-SIT time series for these thicker ice categories for the reasons detailed in Appendix D.

Finally, in Canadian waters, the Baffin Bay, and the Labrador Sea (Figure 7e,f), no decrease in ice area for any category is detected, neither by SMOS-SIT nor by ORAS5. Relative year-to-year variations in ice area also tend to be much smaller than in the other two areas.

The consistency in the time series presented in this section demonstrates that large-scale variability and trend of thin sea ice early in the freezing season can be monitored by both SMOS-SIT and ORAS5 with relative confidence. Both products indicate that year-to-year variability in the pan-Arctic area of thin sea ice is currently strong enough to mask any expected negative trend, and that different regions show distinct –even opposed– variability and trends. These can be related to specific regional anomalies in atmospheric circulation and surface conditions for any given year.

[Figure 7 about here.]

## 6 Discussion

In light of the previously discussed shortcomings and uncertainties both in the current version of the SMOS-SIT data and the current version of the ocean reanalysis, we suggest to proceed with caution. It is clear that there is a generic trend for analysed sea ice to be thicker than what is retrieved from SMOS. Indications are that both problems in the model and in the observations contribute to this.

On the model side, an overly simplistic treatment of open-water sea-ice growth (see Sections 2.2 and 3 and Smedsrud and Martin (2015)) leads to overestimation of ice thicknesses during freeze-up season (October–December). Later in winter, the reanalysis is mostly incapable of simulating the polynyas and fracture zones present in the interior of the ice pack.

On the observational side, low sensitivity of the SMOS brightness temperatures for ice thicknesses larger than 0.5 m is compensated in the SMOS-SIT retrieval algorithm by heavily relying on auxiliary fields from external sources, such as 2 m temperature and winds, sea ice salinity, and snow thickness on sea ice. These have considerable and poorly quantified uncertainties associated with them (e.g. Bauer et al. (2016)), which reflects in uncertainty in the retrieved ice thickness. For ice thicknesses below 0.5 m, the assumption of 100% sea-ice concentration becomes questionable.

The previous example illustrates that reanalysis–observation departures have several distinct root causes, and future data assimilation studies using SMOS should treat each of the following scenarios differently:

1. The model over- or underestimates large-scale ice thickness in the areas of first-year ice. Typical is an overestimation in October–December in the Arctic Shelf Seas. Sea-ice thickness as derived by SMOS is within the range of the unconstrained sea-ice model, so that data assimilation will unequivocally provide a better estimate of the truth than model or observations alone.

2. SMOS-SIT systematically underestimates ice thickness. We argue that this typically occurs in the Baffin Bay and Labrador Sea during late winter. Assimilating SMOS-SIT data here would deteriorate the simulated state. We would argue that the quality of the observational product in this region needs to be improved before using it for data assimilation.

3. SMOS-SIT detects the presence of thin ice in fracture zones and polynyas, but there are fundamental structural deficits in the reanalysis (see discussion in Section 3) that prevent it from simulating these. Here, SMOS-SIT can contribute to model validation and improvement. Assimilating SMOS-SIT data would lead to a better state estimate, but would force the model outside the range of states it would normally occupy. Assimilation is probably beneficial to arrive at better state estimates and initial conditions, but investigation is needed to ensure no undesired unphysical side-effects are triggered during the assimilation.

With further progress in the retrieval algorithms and the modelling for thin sea ice, the distinction between the above three departure scenario might become obsolete, and unqualified use of the data for model validation and data assimilation will become possible, without the need for manual intervention and interpretation. Until then, we suggest to use SMOS-SIT data as a means of detecting the presence of thin sea ice, and design data assimilation studies with the above three departure scenarios in mind.

## 7 Summary and Conclusions

In this study, we carry out an overall assessment of agreement and discrepancies between SMOS-SIT, an observational product of sea-ice thickness derived from L-band radiances, and ORAS5, an ocean reanalysis that does not assimilate the SMOS-SIT data. We start from the premise that neither the observational product nor the reanalysis can be unequivocally trusted to be closer to the truth, because both of them contain systematic errors that are dependent on the region and feature under consideration. Thus, a careful overall assessment of agreements and discrepancies is advisable before using the observational data routinely for model validation, data assimilation, and forecast verification.

We find that SMOS-SIT and ORAS5 are broadly consistent in distinguishing between areas of newly-formed thin sea ice and areas of old thick sea-ice early in the freezing season. This is true regarding the spatial distribution, but also regarding regional and pan-Arctic interannual variability and trends. However, in terms of reanalysis–observation departures, it is evident

that ORAS5 almost always simulates sea-ice thicker than observed in SMOS-SIT. This happens to a greater or lesser degree, and with various unrelated root causes, depending on the region and feature under consideration.

Early in the freezing season (October–December), there is reasonable correspondence between sea-ice thickness from SMOS-SIT and ORAS5, but sea ice is thicker in ORAS5 than in SMOS-SIT. We suggest that this discrepancy is explained to roughly equal parts by known systematic deficiencies in both products: SMOS-SIT underestimates the true ice thickness because it ignores the open-water contribution to L-band emissivity, and ORAS5 overestimates the true sea-ice thickness because of exaggerated ice growth rates due to limitations inherent to the mono-category approach to modelling the sea-ice thickness distribution.

As the freezing season progresses, ice thicknesses are continuously growing in ORAS5 almost everywhere, but are stagnating and often even decreasing in SMOS-SIT. This stagnation and saturation of sea-ice growth in SMOS-SIT occurs even when only considering data that is deemed to be reliable according to the diagnostic uncertainty parameters provided with the product. The result of this are large discrepancies between SMOS-SIT and ORAS5 sea-ice thickness late in the freezing season (February–April) for all regions except the central Arctic and the Barents and Kara Seas. In the central Arctic Ocean (excluding the surrounding marginal Seas), both SMOS-SIT and ORAS5 agree that there is no thin sea ice ice late in the freezing season. In the Barents and Kara Seas, the departures are moderate throughout the freezing season.

The large positive reanalysis–observation departures late in the freezing season fall into two distinct categories. The first category is prevalent in the Laptev, East Siberian, Chukchi, and Beaufort Seas, where extensive refrozen polynyas and fracture zones exist, as evidenced by independent observations from campaigns and visual imagery. These are well detected by SMOS-SIT, but ORAS5 is mostly unable to simulate them. In this case, the discrepancy can be attributed to errors in the model and data assimilation methods. The second category of large positive departures is most apparent in the Baffin Bay: here, SMOS-SIT ice thickness saturates at values around 0.7 m, whereas simple energy budget considerations, ORAS5 as well as independent observations from radar altimetry suggest values closer to 1.5 m. Hence, it seems that SMOS-SIT is systematically biased low in this case. We suggest several plausible hypothesis for the bias, the most appealing being that SMOS-SIT misinterprets the contribution of appreciable area fractions of open water to L-band emissivity.

The discrepancies described above illustrate that a robust and reliable quantification of the thickness of thin sea ice is from L-band observations and ocean reanalysis is an open challenge. Meeting it will require improvements in the observational methods, but also in the forecast model and data assimilation methods. It should be kept in mind that our capacity to observe and model the thickness of thin sea ice on a pan-Arctic scale is less than a decade old, and many improvements are already imminent. In this light, the consistencies that do already exist are encouraging. We hope that the discrepancies described here will provide inspiration and guidance to future in-depth studies addressing current deficiencies of observational, modelling, and data assimilation methods, so that subsequent improvements can unlock the full potential of L-band radiometry for measuring the thickness of thin sea ice and contributing to an improved characterization and prediction of polar regions.

## Appendix A: Changes from the previous SMOS-SIT version

In the previous SMOS-SIT version 2.1, look-up tables were used in the retrieval algorithm to speed up processing. The resulting discretisation leads to substantial retrieval artefacts. As Figure 8 demonstrates, the frequency distribution of retrieved sea ice thickness (SIT) has an unphysical multi-mode structure, with local minima at around 15, 25, 45 and 80 cm. These modes are

very strong, for instance SMOS-SIT has four times more sea ice at 30cm than at 25cm. This artefact could potentially cause major problems in correct geophysical interpretation of the data, and could cause spurious results when using SMOS-SIT for data assimilation. In the current version 3.1 of the data, the problem has been addressed by introducing more entries in the look-up table with a finer spacing. Furthermore, in the process of converting plane-layer ice thickness into heterogeneous mean ice thickness, instead of using a look-up table, a parametrized conversion function is applied, which avoid the abrupt transition

caused by dividing the ice thickness into discrete entries.

[Figure 8 about here.]

## Appendix B: Ambiguities when retrieving sea-ice thickness from SMOS TB

Sea-ice thickness (SIT) retrieved from L-band microwave radiance is limited by penetration depth of the radiation in sea ice. The maximum retrievable ice thickness is reached when the L-band brightness temperature has no useful sensitivity to SIT

any more, or when it is dominated by uncertainty in the ice bulk salinity and temperature (Tian-Kunze et al., 2014). Figure 9 shows that for SMOS-SIT, throughout the data set, there is a strong functional relationship between retrieved SIT and brightness temperature (TB). TB is very sensitive to SIT of up to 50cm or so, but beyond that the slope TB/SIT of the relationship is small, meaning that SIT is only poorly constrained by TB, and auxiliary data become more important to determine the retrieved SIT.

     Unfortunately, for footprints which are partially open water, SMOS-SIT does not take into account the emission of the open

water. As shown in Figure 9 (middle and right), in the range up to 0.5m̃, there is typically a sizeable open water fraction, and there is a linear relationship between ice concentration and SMOS TB. This suggest that SMOS-SIT erroneously ascribes lower TB to thinner ice instead of open water, and hence below 50cm we must expect SMOS to be biased low (see also Tian-Kunze et al. (2014)). However, this might be compensated by the fact that retrievals for sea ice concentration are often also biased low for areas of thin sea (Kwok et al., 2007). For retrieved ice thicknesses above 0.5m̃, the open water fraction is usually low

so does not contribute much to the TB; however, in this range the retrieved thickness is dominated by potentially uncertain assumptions about snow, ice temperature and ice salinity.

[Figure 9 about here.]

## Appendix C: Day-to-day variability

Sea ice thickness at a particular location retrieved from SMOS-SIT varies much more from one day to the next than analysed by

ORAS5. Figure 10 shows that the distribution of daily SIT changes is much broader for SMOS-SIT than for ORAS5. Extreme

daily thickness changes of more than 0.2 m occur around 6% of the time in SMOS-SIT, but less than 1% of the time in ORAS5. These changes can have either thermodynamic causes (ice mass changes) or advective causes (ice is moved in/out of grid cell). A SMOS-SIT grid cell has a width of 12.5km. For reference, an advective change of 0.2 m would require a nearby step change of 0.2 m in the ice thickness, combined with strong winds or ocean currents that are able to move the ice by 12.5 km in a day.

Alternatively, if the change was thermodynamic, a surface heat flux of $700 \tilde{W}m^2$ over that day for the whole 12.5 km grid cell would be required. These extreme conditions should only be expected to occur near the ice edge, and in polynyas and fracture zones, and therefore daily changes of 0.2 m or more should be rare.

[Figure 10 about here.]

Inspection of maps of daily changes reveals that large sea-ice thickness (SIT) changes in SMOS-SIT are not restricted to the

ice edge, polynyas and fracture zones, but occur over extended large-scale areas that correspond to changing synoptic weather patterns. An example is given in Figure 11. On 16 Nov 2015, ice surface temperatures derived by SMOS-SIT were around -15°C in the Laptev Sea and SMOS-derived ice thicknesses ranged between 0.5 and 1 m. The next day, SMOS-derived ice surface temperatures in this region increased by 5 K in a very coherent and homogeneous structure, while brightness temperatures decreased only slightly and with less spatial coherence. The SMOS-derived SIT over the Laptev Sea thinned coherently by

more than 0.2 m in some areas. Given that it is impossible for the ice to change that way in reality, taking into account both thermodynamic and advective forcing, it must be concluded that this wide-spread ice thinning by 0.2 m from one day to the next is an error in the retrieval algorithm: strong changes in the ice surface temperature, in reality caused by synoptic changes, together with unremarkable change in brightness temperatures, are erroneously interpreted as a strong thinning of the ice.

The unrealistic strong day-to-day fluctuations in the SMOS-SIT data are likely due to either errors in the auxiliary fields, or

due to the assumption of a linear temperature profile within the ice. If there are relevant errors in the auxiliary fields, a quick change in the field will lead to a quick change in the retrieved ice thickness that is not realistic. The limits to the validity of the assumption of a linear temperature profile has been investigated in detail by Maaß (2013). They found that, after abrupt changes in the meteorological conditions, the temperature profile within the ice can take several days to adjust. Based on these results, we tentatively suggest that the assumption of the linear temperature profile within the ice is responsible for the unrealistic

day-to-day changes in the SMOS-SIT data.

However, this question can only be answered satisfyingly by further research which has full control both over the SMOS-SIT retrieval model and the auxiliary meteorological and oceanographic fields. Most of these auxiliary fields are the output of complex data assimilation systems, and therefore advanced and well-studied uncertainty estimates are available. It would be a valuable first step towards assimilation of SMOS brightness temperatures for SIT, if the SMOS-SIT retrieval model could

be installed at one of the centres who produces the auxiliary fields, and test sensitivity of the retrieved SIT to their known uncertainties.

[Figure 11 about here.]

## Appendix D:  Representation of thicker ice

When interpreting sea-ice thicknesses of 0.5 m or higher from SMOS-SIT, it is essential to inspect the provided uncertainties. Neglecting to do so easily results in wrong conclusions. As an example, Figure 12 shows sea-ice thickness on a single day (15 Nov 2012) as seen by SMOS-SIT and ORAS5. When considering all data from SMOS-SIT (Figure 12a), a false impression of almost uniformly 1 m thick sea ice throughout the Arctic Ocean is given, which is unrealistic given the well-known fact that the multi-year ice north of Greenland and the Canadian Archipelago is several meters thick, whereas the newly formed first-year ice in the marginal seas of the Arctic Ocean is probably thinner than 1 m. Sea-ice thickness in ORAS5 (Figure 12b) clearly shows the expected structure, in good agreement with other observations and modelling results (Kwok and Cunningham, 2008; Schweiger et al., 2011; Laxon et al., 2013).

Figure 12c shows the corresponding scatter density between SMOS-SIT and ORAS5 sea ice thickness for the freeze-up season 15 Oct - 15 Dec 2012. It is evident that SMOS-SIT, without any filtering, has lots of ice thickness in the 1-1.5 m range, which do not correlate at all with the ORAS5 ice thickness.

[Figure 12 about here.]

*Acknowledgements.*  This work was partly supported by ESA under the contract 4000101703/10/NL/FF/fk. We thank Nina Maaß, Matthias Drusch, Leif T. Pederson, and Nick Hughes for helpful discussions.

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

**List of Figures**

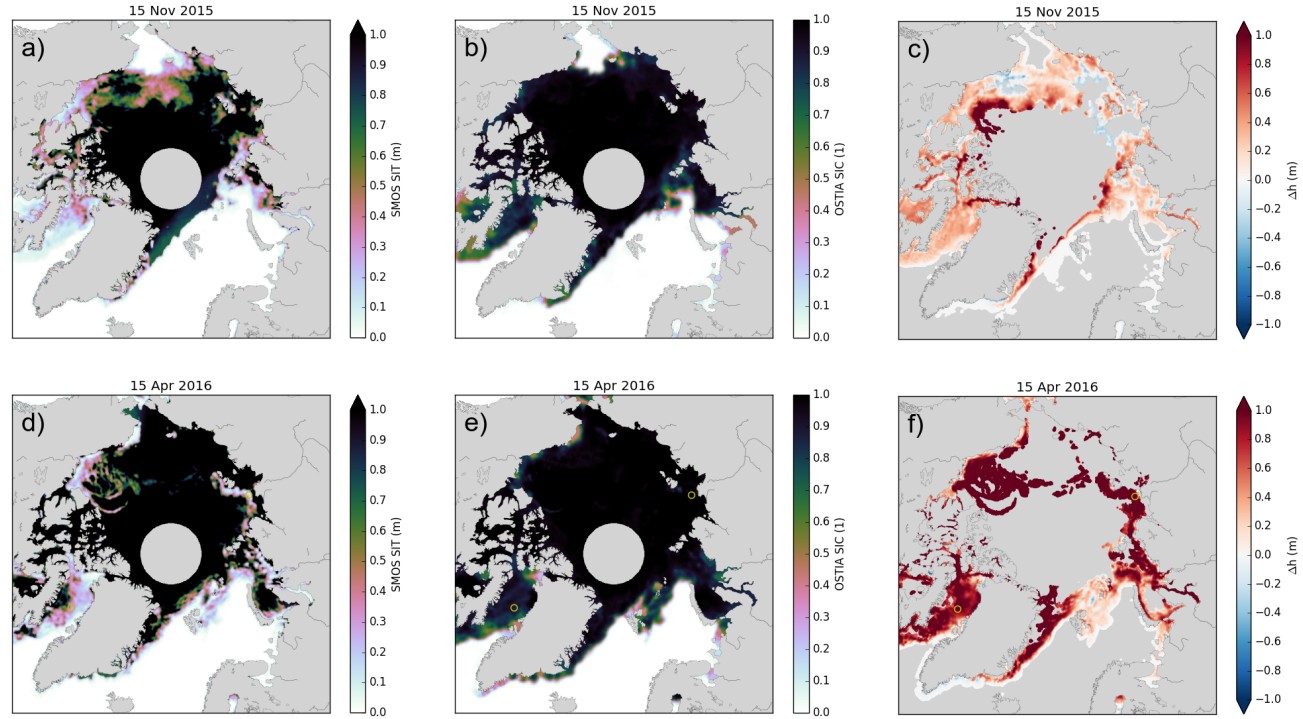

**Figure 1.** Thin sea ice for two selected days representing typical conditions in early and late winter: 15 Nov 2015 (a)–(c) and 15 April 2016 (d)–(f). Subfigures (a) and (d) show the sea ice thickness retrieved by SMOS-SIT. The colours saturate at 1m, because ice thicknesses beyond that can normally not be retrieved. Subfigures (b) and (e) show sea ice concentration from the OSTIA product. The difference between sea ice thickness analysed in ORAS5 and retrieved by SMOS-SIT is shown in (c) and (f). The difference is only shown for data points where the retrieved SMOS-SIT ice thickness is lower than 90% of the maximal retrievable thickness (see Tian-Kunze et al. (2014) for details) and where the SMOS-SIT total retrieval uncertainty is less than 1m. The yellow circles in the Laptev Sea and Baffin Bay in (d)–(f) indicate the representative locations discussed in Section 4.

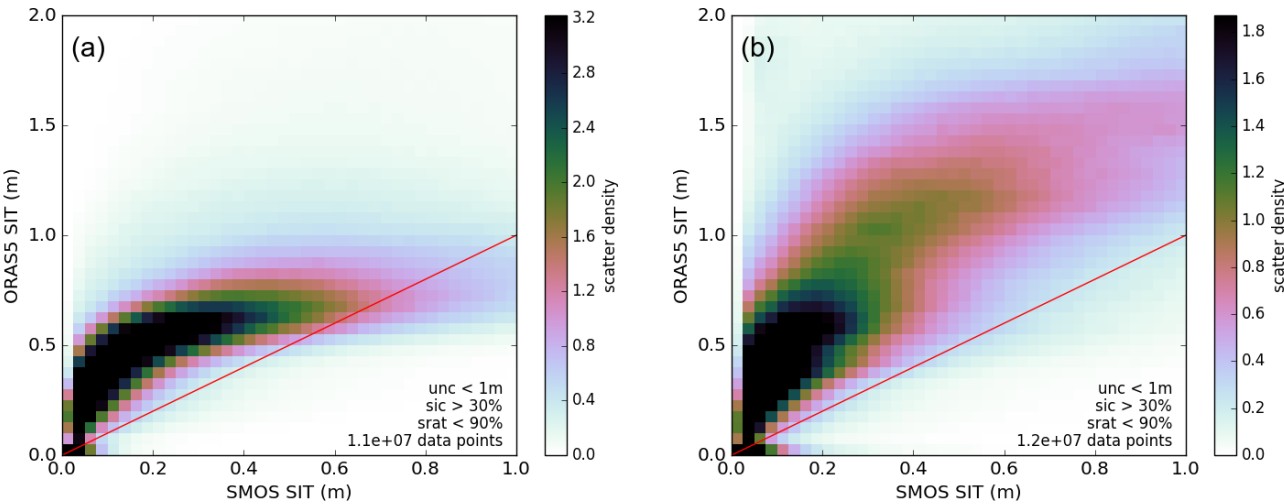

**Figure 2.** Normalized joint frequency distribution (scatter density) of co-located pairs of daily observed and analysed thin sea ice; (a) October to December 2011–2017, (b) February to April 2012–2017. The text insets in the lower-right corner give information on the pre-filtering of the data before producing the scatter density: data points are only considered if the retrieval uncertainty is below 1 m (unc $< 1$ m), the sea-ice concentration from OSTIA is above 30% (sic $> 30\%$ ) and the saturation ratio is below 90% (srat $< 90\%$). The last line of the text inset gives the total number of data points for which the scatter density was calculated.

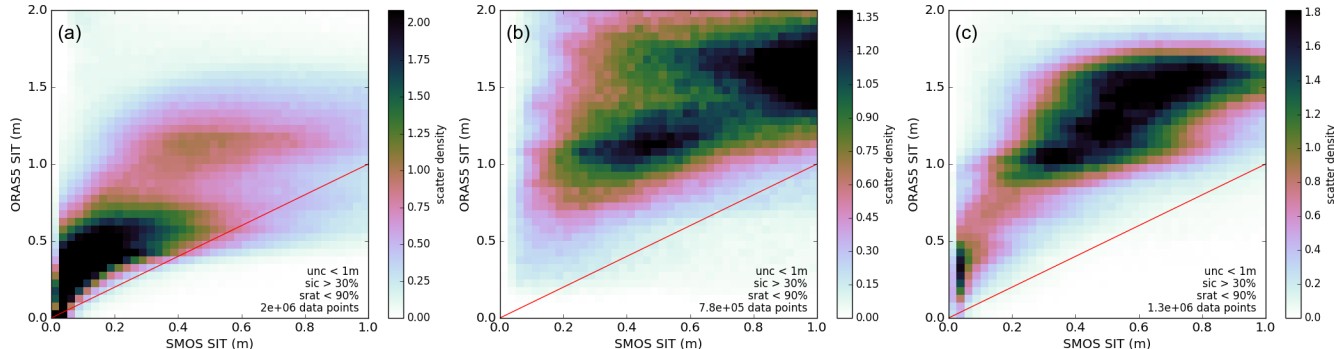

**Figure 3.** Normalized joint frequency distribution (scatter density) of observed and analysed thin sea ice in late winter, February to April 2012–2017: (a) Barents and Kara Seas (15E–90E, 70–85N), (b) Laptev Sea (90E–150E, 70–85N), and (c) Baffin Bay (75W–53W, 65N–80N). For an explanation of the text insets, see caption of Figure 2.

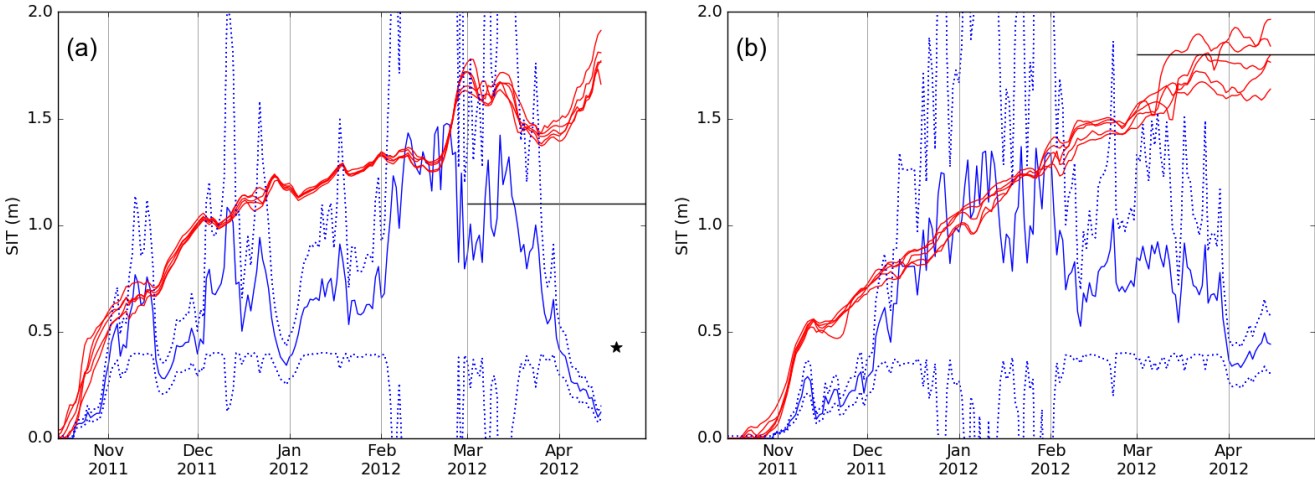

**Figure 4.** Time series of daily sea ice thickness during the 2011/2012 winter at (a) a representative location in the Laptev Sea at 74.5N,127E and (b) a representative location in the Baffin Bay at 72N,62W. Blue is SMOS-SIT (full line) with added and subtracted uncertainty standard deviation (dotted lines); Red are the five realisations of ORAS5; Black horizontal lines are the CryoSat2 average thickness for March/April provided by CPOM; black star is an EM-Bird overfly for the Laptev Sea on 20 April 2012. The corresponding time series of sea-ice concentration are shown in Figure 5b.

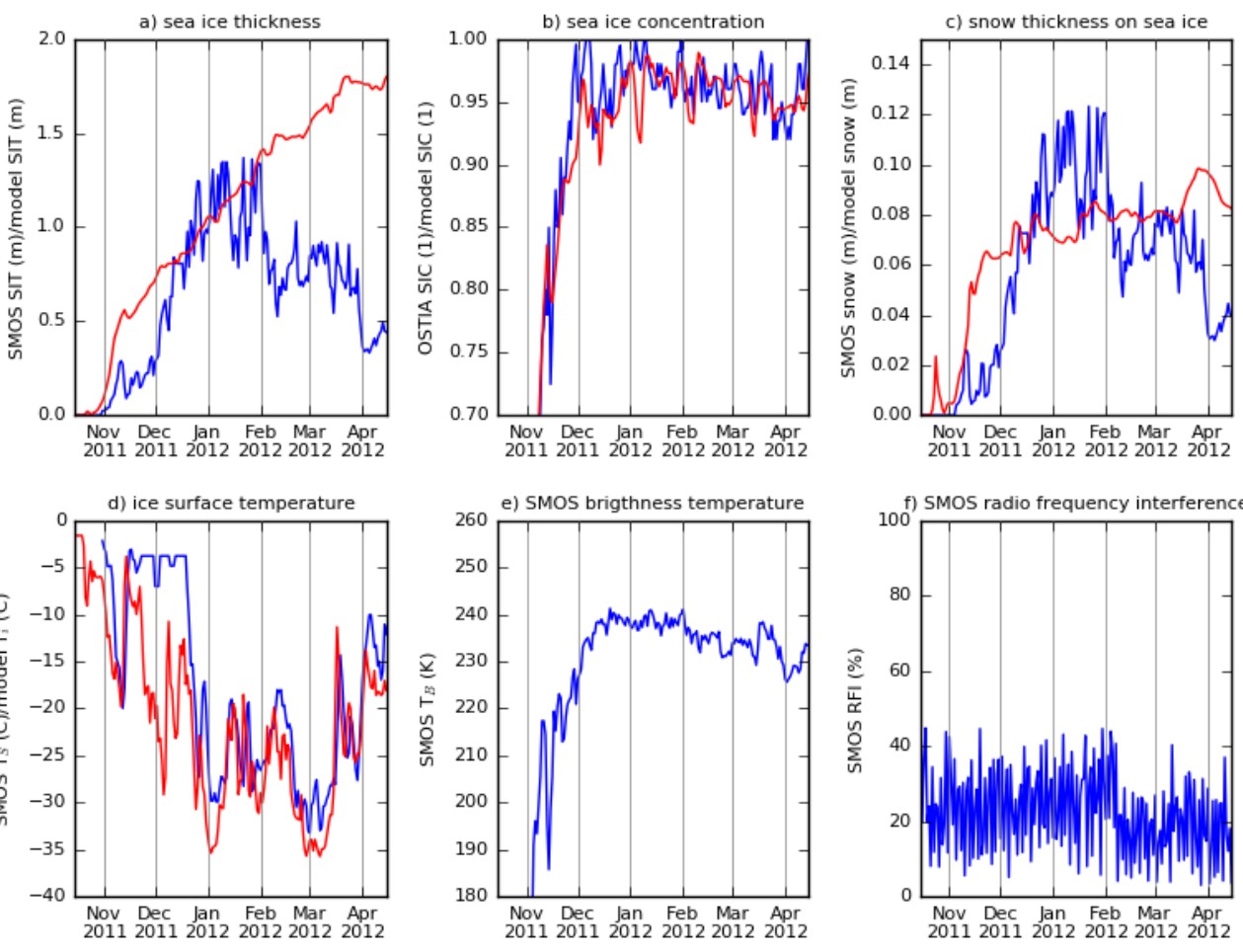

**Figure 5.** Time series for relevant SMOS-SIT and ORAS5 parameters for the Baffin Bay location 72N,62W for the full freezing season 2011/2012. Blue curves are SMOS-SIT parameters (except in (b), where blue is observed ice concentration from OSTIA), red curves are model parameters.

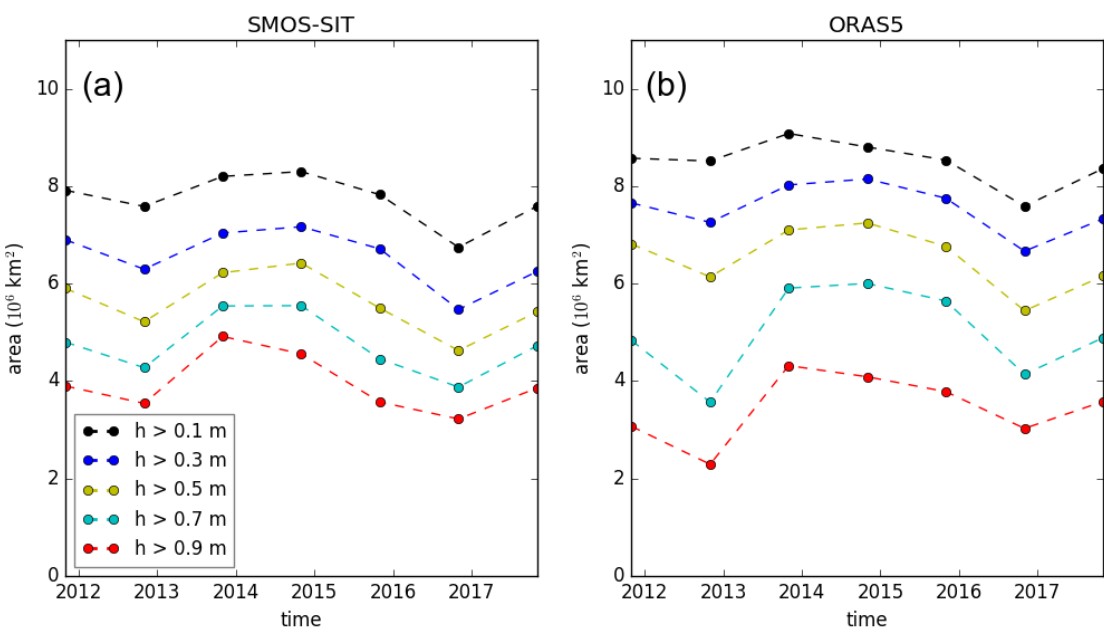

**Figure 6.** Monthly November means of the pan-Arctic area covered by ice thicker than given thresholds in SMOS-SIT (a) and ORAS5 (b).

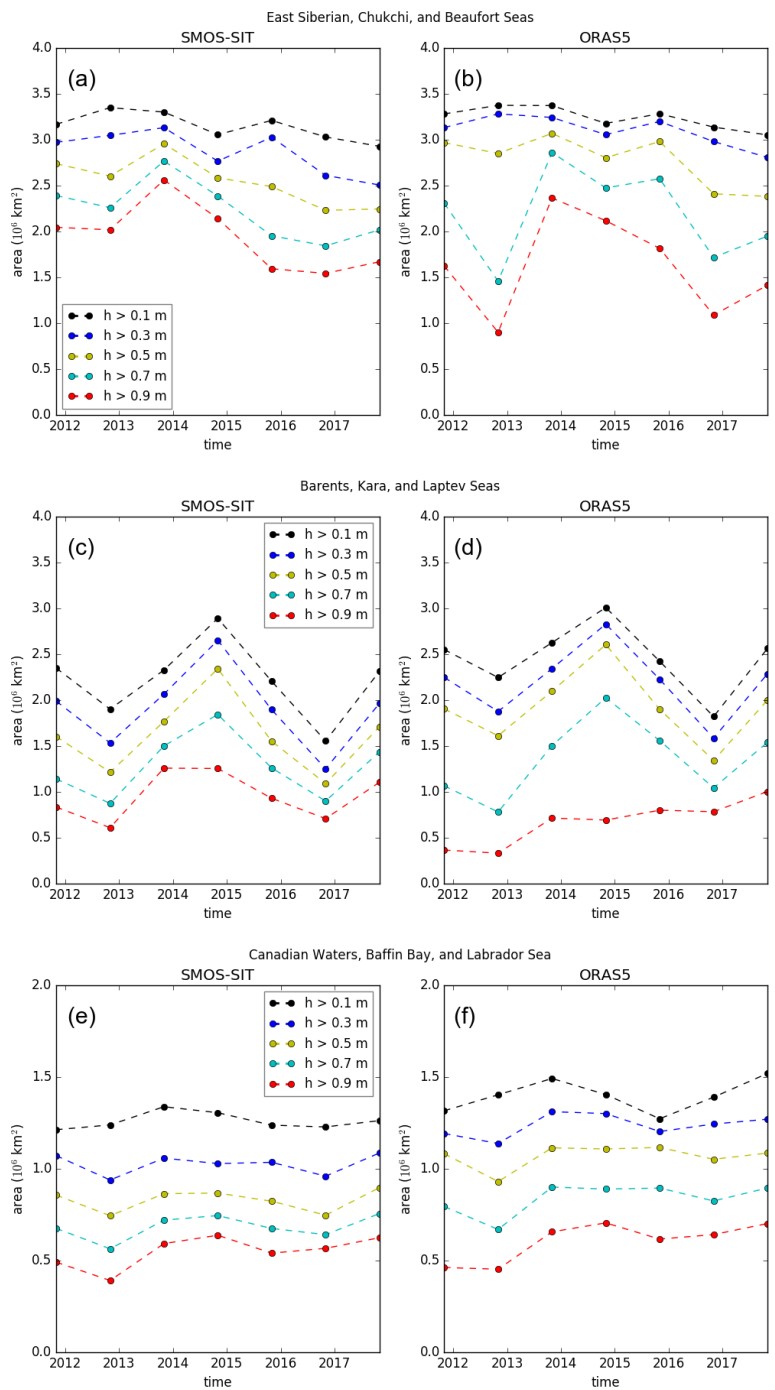

**Figure 7.** Monthly November means of the regional area covered by ice thicker than given thresholds in SMOS-SIT (left) and ORAS5 (right). The boundaries of the longitude-latitude boxes are 0-150E, 70-90N for (a) and (b); 150E-120W, 70-90N for (c) and (d); and 120-70W, 55-83N for (e) and (f).

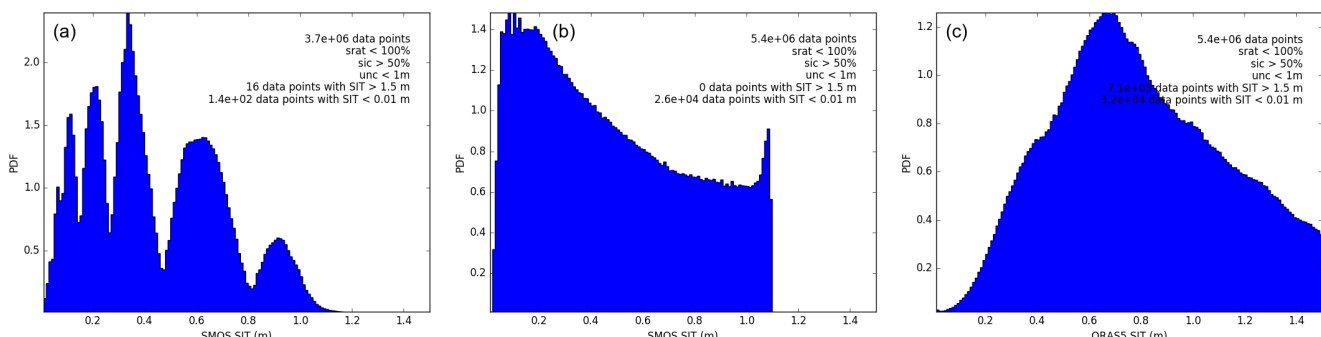

**Figure 8.** SMOS-SIT thickness frequency distribution for the winter 2015/2016 for (a) SMOS-SIT version 2.1, (b) SMOS-SIT version 3.1, and (c) the ORAS5 ocean/sea ice reanalysis.

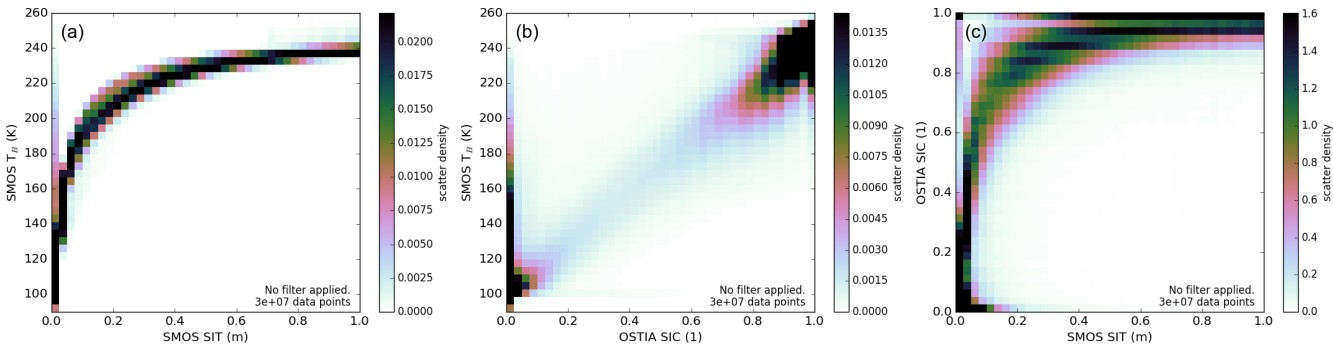

**Figure 9.** Scatter density of (a) SMOS TB and SMOS-SIT-derived sea ice thickness, (b) SMOS TB and sea-ice concentration, (c) sea-ice concentration and SMOS-SIT sea-ice thickness. The scatter density is calculated from all SMOS-SIT data points over the period 15 Oct 2015 to 15 Apr 2016, no filtering has been applied.

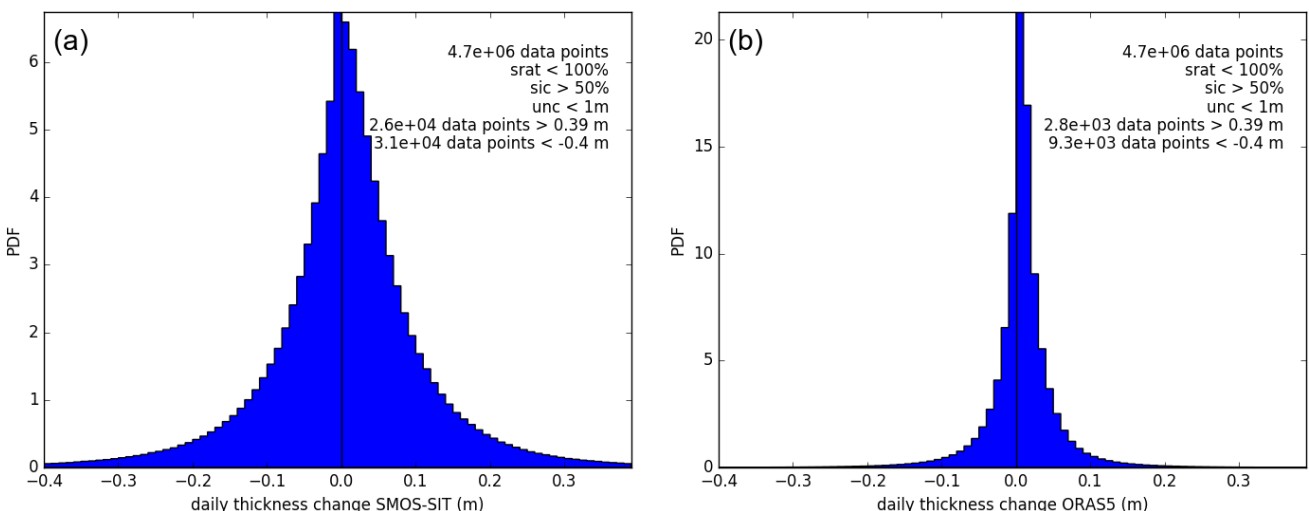

**Figure 10.** Frequency distribution of daily sea ice thickness changes from (a) SMOS-SIT and (b) ORAS5 in the period 15 Oct 2015 to 15 Apr 2016. To produce these histograms, only those differences between consecutive days at the same location have been taken into account where the uncertainty diagnostics provided with SMOS-SIT for both days indicate a reliable retrieval (saturation ration $< 100\%$, uncertainty $< 1$ m, sea-ice concentration $> 50\%$). Day-to-day thickness changes are outside $\pm 0.4$ m in less than 1% of the cases.

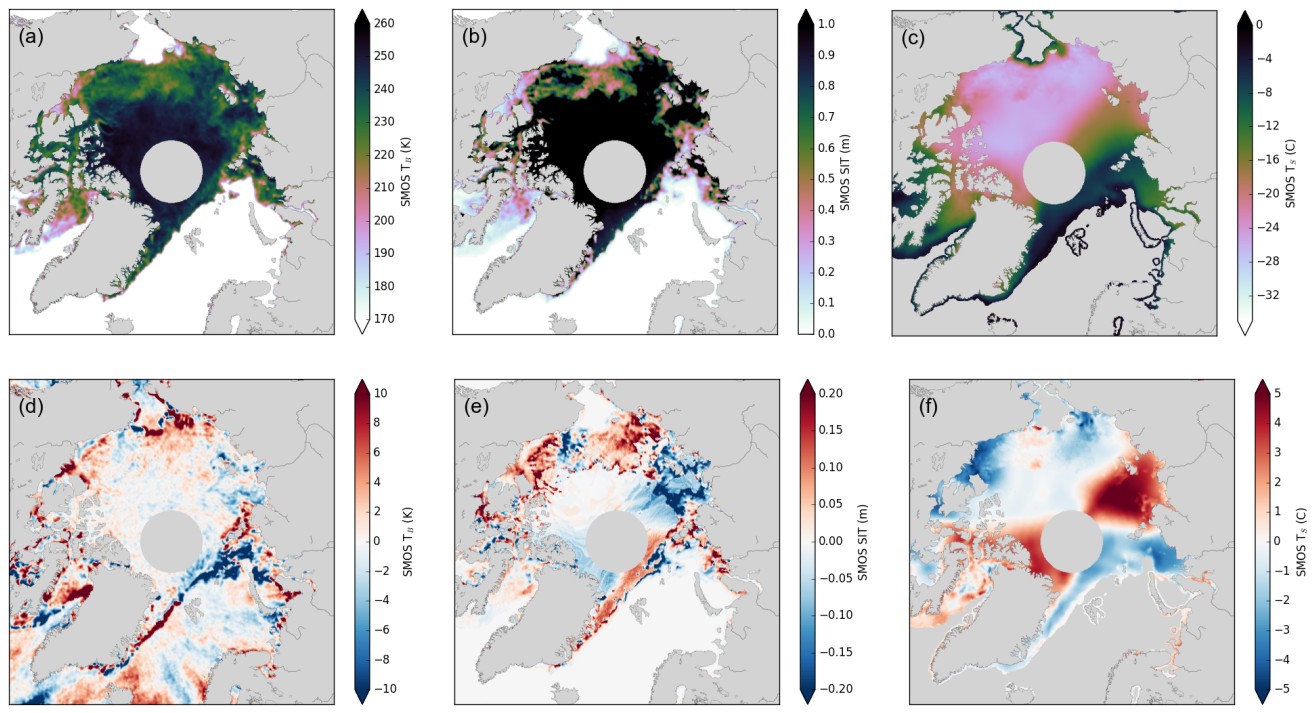

**Figure 11.** SMOS-derived information on 15 November 2015 (top panels) and daily difference between 16 and 15 November 2015 (bottom panels) for SMOS TB (a,d), SMOS-SIT ice thickness (b,e) and SMOS-SIT ice surface temperature (c,f). Correspondence between unrealistic SMOS-derived changes in ice thickness (e) and changes in ice surface temperatures (f) are evident.

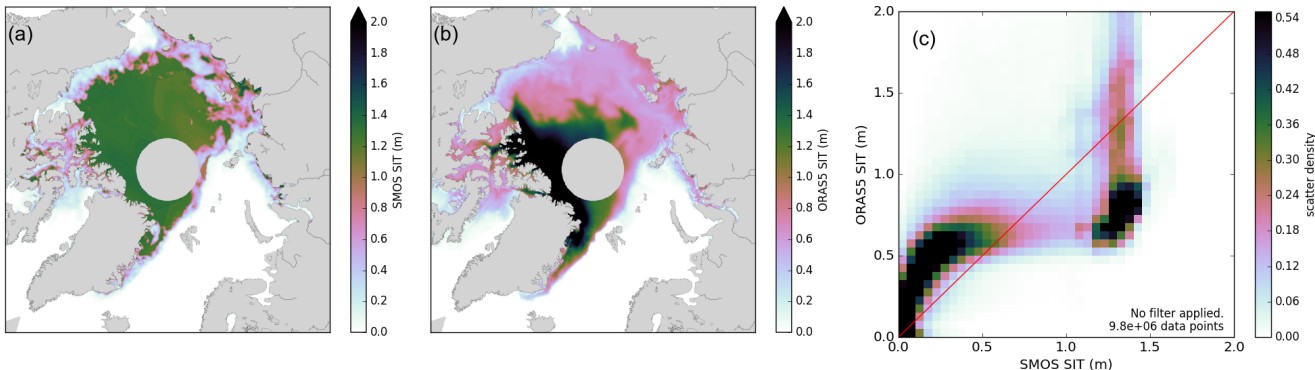

**Figure 12.** Representation of thicker ice in SMOS-SIT and ORAS5. (a) and (b) show sea-ice thickness on 15 Nov 2012 in the range 0–2 m derived from (a) SMOS-SIT and (b) ORAS5. (c) shows the scatter density of ice thickness from SMOS-SIT and ORAS5 for all observation points without any filtering from 15 Oct to 15 Dec 2012.