# Peer review of "Thin Arctic sea ice in L-band observations and an ocean reanalysis"

_The Cryosphere, 2017_

## Referee Comment (RC1) · Anonymous Referee #1 · 19 Dec 2017

The paper provides a useful story on the potential and the pitfalls of using SMOS derived sea ice thickness for the validation and assimilation with an ocean reanalysis. The paper compares SMOS sea ice thickness with ORAS5 reanalysis sea ice thickness. It finds strong correlations, considerable biases and also areas where there is little agreement between SMOS and ORAS5. Some ideas are presented why this disagreement maybe both due to retrieval and modeling errors. While those results are not conclusive, they provide some guidance on how to proceed further and how to potentially incorporate SMOS sea ice information into an ocean reanalysis. I find the paper to be well written and claims sufficiently supported by the evidence. While one may have hoped for some stronger conclusions, I think it is useful as is and provides an incremental contribution.

[Figure]

Specific Comments: Page 3, Line 12... JRA-55.... Later JRA-25 is indicated, please clarify Page 4, Line 33„ Thus the OSTIA ice concentration product can not... I don't understand what is stated here, I must be missing something. Page 5, Line 16, most of this is likely due to the model being unable to simulate the coastal polynya in the Laptev Sea Why is this? I think that could be probed a little more? Is the ice too thick to be advected away from the coast and create the polynya or does it regrow too quickly? Is this a resolution effect? If ice concentrations are assimilated and they show open water there (L-Band does so I assume the higher frequency ice concentration does too?), then why doesn't the model. I understand that this is not necessarily a model validation paper but given the uncertainty in both model and observations it would be good to tie this down a bit more, particularly since later the model seems to be favored over the observations in the case of the Laptev sea. Page 6, Line 18, polynya... as mentioned above, why does the model not show open water areas that SMOS shows and presumably should be visible in the ice concentration data that are assimilated Page 6, Line 28, under the ice This could use a reference Page 7, SST information cannot be used. Again, how come the model doesn't show the open water if it is there in the OSTIA ice concentrations. If there is open water, why can't you assimilate the SST (if they are available). I can't quite follow this argument. I have a sense that this may be an issue with the model which is biased thick and has excessive internal ice strength which keeps the ice from moving off shore. Though this doesn't explain why the assimilation doesn't create the opening. Another plausible explanation might be that excessive ice production due to excessive advection creates too much ice in this area. A look at advection and growth rates in the model might be helpful. This is particularly important since the authors seem to give the Model and Cryosat measurements the upper hand while discounting EM and SMOS measurements. EM measurements aren't really discussed. Page 8, Line 8, Surface Temperature Clarify if ice or air temperatures, I think you mean ice Page 8, Line 10 two reanalysis Correct JRA-25/55 issue see above and remind readers how the JRA reanalysis is used in the SMOS retrievals.
Page 8, Line 26, various thickness classes ORAS5 has thickness classes? I though it was a single category model? Page 10, Line 3, lack of thickness categories in combination with an artificial thickness Please clarify, I can't follow this Page 10, Line 5, incapable of simulating the polynyas Is this because of the lack of thickness categories or a general bias in ice thickness and associated ice strength? How does the model do in general with respect to ice thickness in the interior pack? That information would be useful. Page 10,Line 3 structural limitations Note them please Figure 1. Please explain saturation ratio and where the 90% threshold comes from. Figure 2: Scatter density... what's the unit of density in this context. All scatter plots could use some statistics (e.g. correlation, bias, RMS error in either the figure or caption Fig 4: with added and subtracted.... Add uncertainty Not much discussion is given to the EM data point and why this seems to be rather supporting SMOS than both CryoSat and the Model.

---

## Referee Comment (RC2) · Anonymous Referee #2 · 29 Dec 2017

This manuscript presents a comparison of Arctic sea ice thickness within a range of 0-1 m, both retrieved from SMOS satellite-based L-band brightness temperatures and from a numerical ocean-sea ice reanalysis system assimilating various observational data. It focuses on evaluating regional biases between the two products during the winter 2011-2012 season, but also touches on interannual variations and trends across the full 2011-2016 period. The premise for the study, although unfocused, is valid. Numerical sea ice forecasting systems should unequivocally be more reliable if they can assimilate a greater breadth and variety of observational data, such as low-frequency passive microwave retrievals of ice geophysical properties like those provided by SMOS. Here, the authors appear to be undecided on the main purpose of their study: is the idea to verify/validate the ORAS5 forecasting system using the SMOS data? If so, given

the observational uncertainties discussed in the manuscript, the SMOS data do not appear ready for this. Moreover, why was the enhanced sea ice thickness product incorporating SMOS and Cryosat-2 data not utilized. Alternatively, is the idea to evaluate the root causes of biases within the SMOS data? In which case, this is mostly done qualitatively. Several possible reasons are introduced to explain uncertainties in the SMOS data, but none are investigated in detail so no useful conclusions are made. Given that the premise of validating numerical sea ice forecasting systems is highly valuable, I recommend this paper could be published following major revisions. In line with comments above, the authors should decide exactly what they want the paper to be, to allow them to focus their arguments into quantitative useful conclusions.

General Comments

• Re. Section 2.1, do you have quantitative component uncertainties for each of the contributing factors listed here (e.g. uncertainty contribution from the smos Tb, from the ancilliary T and S data, from using assumptions for linear T-gradient, desalinization scheme etc.)? Are these provided in the SMOS product or can they be provided by the co-authors? In the context of the entire study this would be very useful, as it would allow the authors to better evaluate regional biases in the SMOS data and thus understand how likely identified bias is a product of the SMOS data or forecasting system. An example of where this would be useful is around page 7 line 32.

• It would be valuable to include all or details from Appendix C in the main paper. This extra understanding of where and in what context the SMOS data could be limited would really help to interpret the validity of results from the forecasting system. This analysis could be expanded by examining scales of day-to-day variability between a fast-ice region (e.g. the CAA) and a dynamic region, over the same time period or scenario (like the author's rapid air T change). Equally, more depth to the analysis between ice concentration and smos ice thickness (also in the appendices) and on the effect of ancillary fields on the ice thickness retrievals would be incredibly valuable and relevant, even though the authors suggest this is beyond the scope of the paper.

• Section 5 is quite vague and unfocused. The bulk of the paper would be more useful if this was removed and replaced with more detailed investigation of regional model-obs biases, investigating particular causes for the regional biases the authors touch upon in the previous section.

• You mention at Page 10 line 6 that the smos ice thickness algorithm relies much more on ancillary fields when ice thickness >0.5m. It would therefore be useful to analyse model-obs biases for different categories of uncertainty or for different ice thickness categories. Is there a strong relationship between bias magnitude and smos-sit or uncertainty?

• I do not agree with the statement there is 'reasonable agreement' between observed and analysed ice thickness in the early freezing period. There is systematic nonlinear bias, which has not been explained or properly quantified here.

• To reiterate an earlier point, it is difficult to understand whether the idea of the paper is to verify/validate the reanalysis system (in which case it would have made more sense to use the combined cs-2/smos product from AWI and Hamburg http://data.seaiceportal.de/gallery/index_new.php?active-tab1=measurement&ice-type=thickness&satellite=CS®ion=n&resolution=weekly&minYear=2017&minMonth=4&minDay=3&maxYear=2017&maxtab2=thickness), or to verify/test smos (in which case it is diffcult to use a highly simplified model to do this).

Minor Comments

Page 1 Line 22, 'coverage at a'

P2 L18, requires more specific objectives for the study, beyond simply compare observations with model. What exactly are you trying to achieve here? What exactly will the study provide that is useful for future work?

P5 L28, is ORAS5 SIT <0.3 m impossible? In what situations do you get very thin ice? SIC very low? A 'freeze-up threshold' is referred to later on but should be explained

here.

P5 L34, you need to explain this non-linear dependence here or in the discussion. Clear dependence within the LIM2 ice redistribution function? Or from the single thickness class assumption? Or is this some bias introduced from SMOS?

P6 L23, this is a very qualitative description of the relationship... Can you explain?

P6 L30, where are they assimilated? Outside the ice edge presumably?

P7 L7, There is lower SIC in BB in April, so this could be caused by the SMOS-SIT assumption of total ice concentration within a grid cell? Tb is biased due to the emissivity of open water.

P7 L8, remove 'also' and add appropriate Tilling citation

P7 L22, this is likely owing to low SIC. Linked to the second major point above, some more involved analysis SMOS-SIT sensitivity and higher frequency emissivity/SIC would be very useful and may allow you to make much more robust arguments for causes of obs/model bias.

P8 L5, close to 100%, but not at it, whereas most other regions have total ice concentration. Another thing to consider is that sea ice in BB is fairly low latitude so could be melting some years in April and affecting the L-band penetration depth. What do the PMW data suggest in terms of melt onset date for BB in 2012? Crucially, do you observe this clear bias every year for BB?

P9 L14, change 'than' to 'then'

P10 L16, this would be a much stronger argument if you could provide reasonable evidence as to why this happens. Do you even see the same biases every year? Could you test the interannual persistence of your regional biases? Again this would be highly valuable to the community.

P10 L23, surely more relevant here is the need to improve the rheology and add formulations to the numerical scheme to allow for polynya development, rather than just assimilating observations and the model re-equilibrating to incorrect/overestimated ice thickness?

P12 L18, this is an important limitation that could have been examined in greater detail within the main paper.

P13 L 13, this is a very useful finding that could be represented better in the main paper and given as one of the paper's main conclusions.

Fig 2, explain what 'unc', 'sic' etc. mean within figure caption.

Fig 2, is it impossible to get forecast SIT below 0.3 m when SIC is low (i.e. when smos-sit is around 0)? Why?

Fig 3, does (c) show saturation in the smos-sit signal above approx.. 0.5 m? Plateaus above this value, so no sensitivity from L-band signal?

Fig 3, doing this for only one year's winter enhances the possibility for anomalous ice conditions to explain the departures between observed and predicted IT. What do these look like for multiple years? Your arguments would be more convincing if similar patterns of regional biases were found in several/all years.

Fig 4, Mark on a map either here or on fig 1. Adding a panel of SIC would be very useful for analysis.

Fig 4, 'added and subtracted' what? Uncertainty?

Fig 5c, why does snow depth appear to drop considerably throughout the season?

Fig 5e, ice emissivity masked by overlying snow?

Fig 6, remove 'none'.

---

## Author Comment (AC1) · 22 Mar 2018

ï£ijï£ijWe would like to thank both referees for their careful reading and checking of the manuscript, and for making many thoughtful comments and valuable suggestions that helped us to improve it. We attach a single pdf file that contains our responses to the comments by both referees (RC1 and RC2 of the interactive discussion).

Please also note the supplement to this comment:
https://www.the-cryosphere-discuss.net/tc-2017-247/tc-2017-247-AC1-supplement.pdf
* * *
[Figure]

**Supplement:**

**Authors response to referees comments for manuscript tc-2017-247**

*We would like to thank both reviewers for their careful reading and checking of the manuscript, and for making many thoughtful comments and valuable suggestions that helped us to improve it.*

*The numbering of pages, lines, sections and figures used in this response refers to the* old *version of the manuscript. Throughout our response, we use the following abbreviations:*

- *SIC - sea ice concentration*
- *SIT - sea ice thickness*
- *SST - sea surface temperature*
- *PMR - passive microwave radiometry*
- *TB - (microwave) brigthness temperature*

*The manuscript and reviewer comments are published online in The Cryosphere Discussions at* `https://doi.org/10.5194/tc-2017-247`.

**1 RC1 – comments of first referee**

**1.1 Summary and assessment**

**Comment:** *The paper provides a useful story on the potential and the pitfalls of using SMOS derived sea ice thickness for the validation and assimilation with an ocean reanalysis. The paper compares SMOS sea ice thickness with ORAS5 reanalysis sea ice thickness. It finds strong correlations, considerable biases and also areas where there is little agreement between SMOS and ORAS5. Some ideas are presented why this disagreement maybe both due to retrieval and modeling errors. While those results are not conclusive, they provide some guidance on how to proceed further and how to potentially incorporate SMOS sea ice information into an ocean reanalysis. I find the paper to be well written and claims sufficiently supported by the evidence. While one may have hoped for some stronger conclusions, I think it is useful as is and provides an incremental contribution.*

**Response:** We thank the reviewer for a careful reading and assessment of the manuscript, and for suggesting several changes that helped us to improve it.

**1.2 Specific Comments**

**Comment 1.2.1:** *Page 3, Line 12 "...JRA-55...": Later JRA-25 is indicated, please clarify.*

**Response:** We have clarified this. Whereas previous versions of SMOS-SIT used JRA-25 until 2014 and JRA-55 from 2014 onwards, the version 3.1 that we are discussing uses only JRA-55.

**Comment 1.2.2:** *Page 4, Line 33 "Thus the OSTIA ice concentration product can not...": I dont understand what is stated here, I must be missing something.*

**Response:** We meant to say that SIC from convential PMR cannot be used to distinguish areas of old thick sea ice from areas of new thin sea ice because both often have sea-ice concentration of virtually 100%. In contrast, L-Band radiances can be used to make that distinction. We have rephrased P4 L32f. to make the statement more understandable.

**Comment 1.2.3:** *Page 5, Line 16: Most of this is likely due to the model being unable to simulate the coastal polynya in the Laptev Sea. Why is this? I think that could be probed a little more? Is the ice too thick to be advected away from the coast and create the polynya or does it regrow too quickly? Is this a resolution effect? If ice concentrations are assimilated and they show open water there (L-Band does so, I assume the higher frequency ice concentration does too?), then why doesn't the model. I understand that this is not necessarily a model validation paper but given the uncertainty in both model and observations it would be good to tie this down a bit more, particularly since later the model seems to be favored over the observations in the case of the Laptev Sea.*

**Response:** These are very good questions and suggestions thank you. Before answering, we think it is necessary to clarify two issues mentioned in the reviewer's comment first: (1) As shown very clearly in Figure 1e, SIC in the Laptev Sea polynya is close to 100%, so assimilating SIC does not help and might even be detrimental. In winter, polynyas often refreeze very quickly and are then covered by thin ice. There is a crucial difference in emissivity between higher-frequency microwave and L-Band microwave radiation for the thin ice in the refrozen polynya, which is exactly the point we are trying to make. (2) It seems to be a misunderstanding that "later the model seems to be favored over the observations in the case of the Laptev Sea" – to the contrary! Figs. 3b and 4a and their corresponding discussion in the main text quite clearly argue that the refrozen Laptev Sea polynyas are detected by L-band observations but not simulated by the reanalysis. We have rephrased sentences in the text that could lead to misunderstanding.

This leaves the question why the reanalysis does not simulate the polynya. This is an important question to tackle for model and data assimilation development, but it is not easy to answer because there are many possible reasons, and a dedicated study would be needed to narrow them down and provide a confident answer. In the light of the above, it could even be that the implied SIT increments from the SIC assimilation are responsible. We have added some discussion on this to the text on P5 L16.

**Comment 1.2.4:** *Page 6, Line 18 "polynya...": as mentioned above, why does the model not show open water areas that SMOS shows and presumably should be visible in the ice concentration data that are assimilated?*

**Response:** See our reply to the previous comment. It is evident from higher-frequency PMR that the areas we are referring to are are covered by thin ice. We acknowledge that a polynya in the strict sense is an area of open water surrounded by ice, and our usage of the term might therefore be misleading. We went through the entire text of the manuscript and added clarification that we are talking about refrozen polynyas (e.g. P5 L11 and L16, P6 L18, and others)

**Comment 1.2.5:** *Page 6, Line 28 "under the ice": This could use a reference to page 7, SST information cannot be used. Again, how come the model doesn't show the open water if it is there in the OSTIA ice concentrations. If there is open water, why cant you assimilate the SST (if they are available). I cant quite follow this argument. I have a sense that this may be an issue with the model which is biased thick and has excessive*

*internal ice strength which keeps the ice from moving off shore. Though this doesn't explain why the assimilation doesn't create the opening. Another plausible explanation might be that excessive ice production due to excessive advection creates too much ice in this area. A look at advection and growth rates in the model might be helpful. This is particularly important since the authors seem to give the model and CryoSat measurements the upper hand while discounting EM and SMOS measurements. EM measurements aren't really discussed.*

**Response:** We have added a reference to using SST information on P6 L28. As explained in our responses to the previous comments, there is no open water in the frozen polynyas and hence SIC assimilation does not help. We agree it would be very interesting to investigate why the reanalysis does not represent the frozen-over polynyas, but as argued in our response to comment 1.2.3 we think this is a study in its own right and out of the scope of this manuscript. We have added discussion on the points mentioned by the reviewer on P5 L18.

**Comment 1.2.6:** *Page 8, Line 8: "Surface Temperature": Clarify if ice or air temperatures, I think you mean ice.*

**Response:** We did indeed mean the ice surface temperature. We have rephrased that sentence to make it clearer.

**Comment 1.2.7:** *Page 8, Line 10 "two reanalysis": Correct JRA-25/55 issue see above and remind readers how the JRA reanalysis is used in the SMOS retrievals.*

**Response:** Revised as suggested.

**Comment 1.2.8:** *Page 8, Line 26 "various thickness classes": ORAS5 has thickness classes? I though it was a single category model?*

**Response:** Here, we refer to the diagnostic thickness classes we defined for producing the figure. It is unfortunate that this can be confused with prognostic thickness classes in the sea ice model. We have revised the sentence and use the term "thickness threshold" to avoid this ambiguity.

**Comment 1.2.9:** *Page 10, Line 3 "lack of thickness categories in combination with an artificial thickness": Please clarify, I cant follow this.*

**Response:** We have revised this sentence and provide a reference to the Section 2.1 of the main text, where we have added a detailed explanation of this issue. We have also added an appropriate literature reference.

**Comment 1.2.10:** *Page 10, Line 5 "incapable of simulating the polynyas": Is this because of the lack of thickness categories or a general bias in ice thickness and associated ice strength? How does the model do in general with respect to ice thickness in the interior pack? That information would be useful.*

**Response:** This is a recurring comment, we refer to our answer to comment 1.2.3. Regarding the general bias in ice thickness, we point out that the model does well relative to its peers, as shown in Uotila et al. (2018). We have added this reference to the model description Section 2.2.

**Comment 1.2.11:** *Page 10, Line 20 "structural limitations": Note them please.*

**Response:** In response to this and other comments, we have added a paragraph in Section 2.2 that explains the simplified treatment of thin ice in the sea-ice model and provides relevant references to the literature. Other structural limitations might be less

obvious and require further research and experimentation to corroborate, so we cannot note them here. We have rephrased the sentence on P10 L20, and have provided a reference to Section 3, where we discuss potential structural limitations.

**Comment 1.2.12:** *Figure 1: Please explain saturation ratio and where the 90% threshold comes from.*

**Response:** If an ice thickness change of $1\,\mathrm{cm}$ leads to a TB change of less than $0.1\,\mathrm{K}$ in the SMOS-SIT retrieval algorithm, the TB is considered saturated. The ice thickness at which that happens for the current values of the auxiliary fields is the maximal retrievable ice thickness $d_{\max}$. The saturation ratio of any other retrieved ice thickness $d$ for the same values of the auxiliary fields can then be expressed as $d/d_{\max}$ (Tian-Kunze et al., 2014). We have added this explanation to P3 L19ff. of the main text, and we have reworded the caption of Figure 1.

**Comment 1.2.13:** *Figure 2: "Scatter density...": Whats the unit of density in this context. All scatter plots could use some statistics (e.g. correlation, bias, RMS error) in either the figure or caption.*

**Response:** We use scatter density as a synonym for "normalized bivariate joint frequency distribution", so it has no units. We have added an explanation to the main text on P5 L24 and have reworded the figure caption to improve clarity.

**Comment 1.2.14:** *Fig 4: "with added and subtracted...": Add uncertainty. Not much discussion is given to the EM data point and why this seems to be rather supporting SMOS than both CryoSat and the Model.*

**Response:** Thank you for spotting the omission of "uncertainty". There is no discussion on the EM data point because we consider it to be a much better estimate of the truth than any satellite-derived observation or model simulation. The fact that it supports SMOS much better than the model is exactly the point we are trying to make (see several previous comments and our responses): The re-frozen polynyas are real, and they are detected by SMOS, but not simulated by the model. We have rephrased several bits of text to make this point even more clearly. The mismatch to CryoSat is a different topic and should be subject to further research. In this case, please note that the CryoSat value represents a full month of data and hence cannot be directly compared to a daily snapshot from an EM-bird overflight and a SMOS-SIT daily mean.

**2 RC2 – comments of second referee**

**2.1 Summary and assessment**

**Comment 2.1.1:** *This manuscript presents a comparison of Arctic sea ice thickness within a range of $0-1\,\mathrm{m}$, both retrieved from SMOS satellite-based L-band brightness temperatures and from a numerical ocean-sea ice reanalysis system assimilating various observational data. It focuses on evaluating regional biases between the two products during the winter 2011-2012 season, but also touches on interannual variations and trends across the full 2011-2016 period. The premise for the study, although unfocused, is valid. Numerical sea ice forecasting systems should unequivocally be more reliable if they can assimilate a greater breadth and variety of observational data, such as low-frequency passive microwave retrievals of ice geophysical properties like those provided by SMOS.*

*Here, the authors appear to be undecided on the main purpose of their study: is the idea to verify/validate the ORAS5 forecasting system using the SMOS data? If so, given the observational uncertainties discussed in the manuscript, the SMOS data do*

*not appear ready for this. Moreover, why was the enhanced sea ice thickness product incorporating SMOS and Cryosat-2 data not utilized. Alternatively, is the idea to evaluate the root causes of biases within the SMOS data? In which case, this is mostly done qualitatively. Several possible reasons are introduced to explain uncertainties in the SMOS data, but none are investigated in detail so no useful conclusions are made. Given that the premise of validating numerical sea ice forecasting systems is highly valuable, I recommend this paper could be published following major revisions. In line with comments above, the authors should decide exactly what they want the paper to be, to allow them to focus their arguments into quantitative useful conclusions.*

**Response:** We thank the reviewer for their careful reading of the manuscript, for pointing out its weak points, and for making several very valuable suggestions how to improve it.

The main purpose of the study is to give an overall assessment of agreement and discrepancy between an observational product of sea-ice thickness from L-band PMR and a sea-ice ocean analysis system that does not assimilate the observational product. Observing and analyzing thin sea ice has only become possible in the last few years, so as with every new technology, initial problems are to be expected. To our knowledge, a detailed assessment of this kind covering the whole Arctic has never been done, yet we see it as an essential step to take before using the observational product for model validation, data assimilation, or forecast verification.

Perhaps we did not make our premise clear enough in the abstract and in the introduction: we do not and will never know the true ice thickness with the vast temporal and spatial coverage provided by remote sensing and reanalysis products. Both can have large errors, and it is not a-priori clear that one is superior to the other. In fact, one of the main points of the manuscript is that discrepancies between the two products compared can be attributed to errors in one or the other, depending on the region and feature considered. Hence, in many cases the fidelity of the SMOS-SIT product is not currently high enough to validate the ORAS5 reanalysis system, as pointed out by the reviewer.

We enthusiastically agree that an investigation of the root causes of potential biases in the SMOS-SIT data and the reanalysis is needed. However, this is *not* the point of this manuscript. As the reviewer points out, we offer possible reasons but can not follow up on them. Numerical experimentation with the retrieval algorithm and the ocean analysis system is beyond the scope of our study. Rather, our study provides concrete examples of discrepancies and so can provide inspiration and guidance for a future study on sensitivities and uncertainties of the retrieval algorithm and the reanalysis.

We have revised the manuscript in order to address the very valid points raised by the reviewer. We think that the new manuscript version does better in presenting the premise and purpose of the study (and thus managing the expectation of the reader), provides some deeper analysis as requested by the reviewer, and summarizes the main points of the paper in the conclusions section more pointedly. The revisions are described in more detail in our responses to the following comments.

**2.2   General Comments**

*Comment 2.2.1: Regarding Section 2.1, do you have quantitative component uncertainties for each of the contributing factors listed here (e.g. uncertainty contribution from the smos Tb, from the ancilliary T and S data, from using assumptions for linear T-gradient, desalinization scheme etc.)? Are these provided in the SMOS product or can they be provided by the co-authors? In the context of the entire study this would be very useful, as it would allow the authors to better evaluate regional biases in the SMOS data and thus understand how likely identified bias is a product of the SMOS*

*data or forecasting system. An example of where this would be useful is around page 7 line 32.*

**Response:** Having a look at quantitative component uncertainties for SMOS-SIT is an excellent suggestion, and we agree it would be extremely useful. They are not currently available from the published SMOS-SIT product, and although they could be provided in principle, this is a non-trivial exercise, both conceptionally and computationally. An ongoing project is investigating this at the moment, and results should be left to a dedicated study which can build on this manuscript for inspiration and guidance. We have added this premise to the introduction, and have also added references to Maaß (2013) who have investigated these uncertainties/sensitivities of the retrieval model for idealized cases, and Richter et al. (2016) who perform an intercomparison of L-Band brightness temperatures calculated from reanalysis sea-ice fields.

**Comment 2.2.2:** *It would be valuable to include all or details from Appendix C in the main paper. This extra understanding of where and in what context the SMOS data could be limited would really help to interpret the validity of results from the forecasting system. This analysis could be expanded by examining scales of day-to-day variability between a fast-ice region (e.g. the Canadian Arctic Archipelago) and a dynamic region, over the same time period or scenario (like the authors rapid air T change). Equally, more depth to the analysis between ice concentration and SMOS ice thickness (also in the appendices) and on the effect of auxiliary fields on the ice thickness retrievals would be incredibly valuable and relevant, even though the authors suggest this is beyond the scope of the paper.*

**Response:** We agree that these points would be extremely valuable to investigate. However, as we have argued in our response to comment 2.1.1, we think this is better left to a dedicated study on the uncertainties and sensitivities of the retrieval model. This requires non-trivial work, as the retrieval algorithm needs to be run many times with systematic and realistic variation to the thermodynamic sea-ice model, auxiliary fields, and brightness temperatures, and possibly employing different radiative transfer models as well (coherent vs. incoherent etc.).

Given that the magnitude of the unphysical day-to-day changes discussed in Appendix C is well within the uncertainty estimate provided by SMOS-SIT, it might be a bit unfair to assign too much emphasis on them. Rather, it illustrates the fundamental need to complement remote sensing observations with physical constraints from a forecast model background in the framework of data assimilation.

**Comment 2.2.3:** *Section 5 is quite vague and unfocused. The bulk of the paper would be more useful if this was removed and replaced with more detailed investigation of regional model-obs biases, investigating particular causes for the regional biases the authors touch upon in the previous section.*

**Response:** We would like to keep this section, because it provides a positive outlook on how variability and change of thin sea ice in the Arctic can be monitored using SMOS-SIT and ORAS5, despite all their discrepancies. This positive message is one of the main points of the paper.

**Comment 2.2.4:** *You mention at Page 10 line 6 that the SMOS ice thickness algorithm relies much more on auxiliary fields when ice thickness $> 0.5\,\mathrm{m}$. It would therefore be useful to analyse model-obs biases for different categories of uncertainty or for different ice thickness categories. Is there a strong relationship between bias magnitude and SMOS-SIT or uncertainty?*

**Response:** The dependence of the departures on the retrieved ice thickness in SMOS-

SIT can be read off Figures 2 and 3, and we discuss the dependence in the main text. There is no apparent dependence of departures on uncertainty (see Figure 3 in this response), so we have decided not to include it in the manuscript.

**Comment 2.2.5:** *Page 11 line 2f.: I do not agree with the statement that there is "reasonable agreement" between observed and analysed ice thickness in the early freezing period. There is systematic nonlinear bias, which has not been explained or properly quantified here.*

**Response:** We agree there is systematic discrepancy even early in the freezing period. The agreement is "reasonable" only in comparison to the much larger discrepancy later in the freezing season. We have changed the wording on P11 L2. We have added discussion of this nonlinear bias to the main text after P5 L34.

**Comment 2.2.6:** *To reiterate an earlier point, it is difficult to understand whether the idea of the paper is to verify/validate the reanalysis system (in which case it would have made more sense to use the combined CS2/SMOS product from AWI and Hamburg `http://data.seaiceportal.de/gallery/index_new.php?active-tab1=measurement&icetype=thickness&satellite=CS®ion=n&resolution=weekly&minYear=2017&minMonth=4&minDay=3&maxYear=2017&maxMonth=4&maxDay=9&showMaps=y&dateRepeat=n&submit2=display&lang=en_US&activetab2=thickness`), or to verify/test SMOS (in which case it is difficult to use a highly simplified model to do this).*

**Response:** As argued in our response to comment 2.1.1, the premise of the paper is that *both* the current versions of the observational ice thickness product *and* the reanalysis product contain substantial and systematic errors. Hence, careful additional investigation and expert judgement is needed if one wants to use one of them to verify or validate the other. What can be done is to contrast them, and to use independent data and process understanding to give indication as to which of the two is probably closer to the truth for certain identified features and regimes. This is the essence of the paper. We have revised abstract, introduction and conclusions of the manuscript to clarify this point.

Regarding the suggestion to use the combined CS2SMOS product, we note that problems in a multi-sensor product like CS2SMOS are even more difficult to track down. The CS2SMOS ice thickness might be closer to the truth than SMOS-SIT alone, but at the cost of traceability. Besides, our motivation is the potential use of SMOS-SIT for data assimilation. Operational centers are extremely unlikely to assimilate a multi-sensor SIT product, which in itself already is an analysis – it is much preferrable to use products individually and let the analysis system find the best fit to observational data from different sources, that can be inconsistent between themselves.

We have revised the introduction and the conclusions to explain the purpose and scope of the paper better, and to better communicate the main conclusions.

**2.3   Minor Comments**

**Comment 2.3.1:** *Page 1 Line 22: "coverage at a"*
**Response:** Fixed.

**Comment 2.3.2:** *P2 L18 requires more specific objectives for the study, beyond simply compare observations with model. What exactly are you trying to achieve here? What exactly will the study provide that is useful for future work?*
**Response:** This is a valid point and urgently needed to give the right premise for the manuscript. We have revised the introduction to address that (see also responses to the

reviewer's general assessment and general comments).

**Comment 2.3.3:** *P5 L28: is ORAS5 SIT < 0.3m impossible? In what situations do you get very thin ice? SIC very low? A "freeze-up threshold" is referred to later on but should be explained here.*

**Response:** Here and in several other comments the simplified treatment of thin ice in the model is addressed. As alluded to by the reviewer, LIM2 has a minimal floe (or in-situ) ice thickness – new ice will grow at this thickness. This is the "freeze-up threshold" that we are referring to. However, throughout the entire manuscript we compare the *grid-cell mean* ice thickness of the model with SMOS-SIT, because SMOS-SIT also gives the *mean* ice thickness. The thickness at which new ice forms is set to 0.6 m in ORAS5, so a mean thickness of 0.3 m corresponds to exactly 50% area coverage. Mean ice thicknesses below that do exist but are not as abundant (see Figure 1). We have added a sentence on P5 L28 to explicitly state that we compare the grid cell mean ice thickness from both SMOS-SIT and ORAS5. We have also revised added text after P5 L34 that properly explains the "freeze-up threshold" and puts it into context.

**Comment 2.3.4:** *P5 L34, you need to explain this non-linear dependence here or in the discussion. Clear dependence within the LIM2 ice redistribution function? Or from the single thickness class assumption? Or is this some bias introduced from SMOS?*

**Response:** We agree this needed more explanation. We have done some further analysis, with the result that both model and observation deficits mentioned on P5 L5–9 are likely to be important. We have added these results to the text, after the paragraph starting on P5 L26.

**Comment 2.3.5:** *P6 L23, this is a very qualitative description of the relationship. . . Can you explain?*

**Response:** We do not think that this is a qualitative, it is just putting in words what can be seen in the figure. The term "functional relationship" might be poorly chosen. We mean to say that there is a high rank correlation between the two variables (product correlation could still be low due to non-linearity). This can be exploited for a-posteriori calibration. We have reworded these sentences to clarify, referring to the rank correlation instead of a "functional relationship".

**Comment 2.3.6:** *P6 L30, where are they assimilated? Outside the ice edge presumably?*

**Response:** Correct. No SST observations are assimilated in the presence of sea ice for the simple reason that the presence of sea ice makes a satellite observation of SST virtually impossible.

**Comment 2.3.7:** *P7 L7, There is lower SIC in Baffin Bay in April, so this could be caused by the SMOS-SIT assumption of total ice concentration within a grid cell? TB is biased due to the emissivity of open water.*

**Response:** This is an intriguing hypothesis that we had considered at an earlier stage of investigation but then dropped, assuming instead that the real ice cover is 100%. The intrinsic uncertainty of sea-ice concentration from PMR is a few percent even in optimal cases (Ivanova et al., 2015), and if these few percent dominate the L-Band emissivity this invalidates the SMOS-SIT retrieval assumptions. Assuming the TB is 240 K for thick sea ice and 90 K for open water, a simple calculation shows that every percent of open water in a previously closed ice pack will lower L-Band TB by 1.5 K. In the case of the Baffin Bay shown in Figure 5, the SMOS-SIT retrieved ice thickness decreases from

1 m in January to 0.5 m in April while the SMOS TB decrease from 240 K to 230 K and the PMR SIC from close to 100% to about 95% (albeit noisy).

Thus, it is plausible that SMOS-SIT has very low mean sea-ice thickness in late winter in the Baffin Bay because it misinterprets the open-water L-Band signature. This can in principle be tested by restricting to cases where SIC is 100% with high confidence (e.g. where sea ice velocities are convergent or where MODIS visual imagery is available). We have added this hypothesis to the text.

Testing this hypothesis rigorously is outside the scope of this manuscript. However, we can get some indication by plotting the normalized joint frequency distribution (scatter density) of OSTIA SIC and SMOS-SIT SIT for the Western Baffin Bay. Figure 4 shows that there is moderate correlation between SIC and SIT, indicating that the open-water contribution to L-band emissivity matters, but does not dominate the signal.

We have added some discussion on this to the manuscript on P7.

**Comment 2.3.8**: *P7 L8, remove also and add appropriate Tilling citation.*
**Response:** Done.

**Comment 2.3.9**: *P7 L22, this is likely owing to low SIC. Linked to the second major point above, some more involved analysis SMOS-SIT sensitivity and higher frequency emissivity/SIC would be very useful and may allow you to make much more robust arguments for causes of obs/model bias.*
**Response:** This relates to comment 2.3.7. See our response there. The SMOS-SIT sensitivity to open water is not testable given the 100% cover assumption built into the current version of the retrieval algorithm, but it can be seen that it is large by simple back-of-the-envelope calculations (our response to comment 2.3.7, also see Richter et al. (2016)). We have revised P7 of the manuscript to include some discussion on the SIC-sensitivity of L-band PMR as suggested by the reviewer.

**Comment 2.3.10**: *P8 L5, close to 100%, but not at it, whereas most other regions have total ice concentration. Another thing to consider is that sea ice in Baffin Bay is fairly low latitude so could be melting some years in April and affecting the L-band penetration depth. What do the PMR data suggest in terms of melt onset date for Baffin Bay in 2012? Crucially, do you observe this clear bias every year for Baffin Bay?*
**Response:** Agreed, SIC even a few percent lower than 100% will have an important impact on L-band TB. We have revised the text (see response to comment 2.3.7). The second hypothesis of surface melt can be safely rejected for this case, as ice surface temperatures are well below freezing throughout (see Figure 5d in the manuscript). However, it might play a role in other winters.

We have followed the advice of the reviewer to produce the time series for *all* winters, and we have also calculated them for a spatial average over the Western Baffin bay area as defined by Landy et al. (2017), in order to reduce spatial sampling uncertainty. The result is that the behaviour documented in Figure 4 of the manuscript appears in all winters for the entire Western Baffin Bay (Figure 2 in this response).

**Comment 2.3.11**: *P9 L14, change "than" to "then".*
**Response:** Done.

**Comment 2.3.12**: *P10 L16, this would be a much stronger argument if you could provide reasonable evidence as to why this happens. Do you even see the same biases every year? Could you test the interannual persistence of your regional biases? Again*

*this would be highly valuable to the community.*

**Response:** The point of the discussion on P10 is not to claim that there *is* systematic underestimation of sea-ice thickness by SMOS-SIT, but to describe the appropriate action to take in the *scenario* that this is the case (see P10 L10).

However, we can confidently demonstrate that these regional biases robustly occur each year (see our response to several previous comments, e.g. 2.3.10). We have added this to the manuscript, by reproducing Figures 2 and 3 for *all* years available, and by plotting the time series in Figure 4 for all years available and as an area average over the western Baffin Bay (Figure 2 in this response).

**Comment 2.3.13:** *P10 L23, surely more relevant here is the need to improve the rheology and add formulations to the numerical scheme to allow for polynya development, rather than just assimilating observations and the model re-equilibrating to incorrect/overestimated ice thickness?*

**Response:** We agree, it is much preferrable to remove the model bias rather than forcing the model out of its natural state by data assimilation. However, in practice model and data assimilation developments are often not well synchronized, so that data assimilation does correct for model biases. In most cases, assimilating in the presence of model bias is still preferrable to not assimilating, because it leads to better time-evolving state estimates, and because forecasts are improved at least for short lead times when the model has not had time to re-develop the bias.

**Comment 2.3.14:** *P12 L18, this is an important limitation that could have been examined in greater detail within the main paper.*

**Response:** Agreed. We have added more discussion on that to the main text, also in response to comments 2.3.7 and others.

**Comment 2.3.15:** *P13 L13, this is a very useful finding that could be represented better in the main paper and given as one of the papers main conclusions.*

**Response:** This comment ties into the general comment 2.2.1, see our response there. We agree that this is an important aspect, but it is impossible to draw useful quantitative conclusions on this from a purely diagnostic point of view (which is what we do in this paper). We think it can only be a strong conclusion in a study that explicitly changes parameters of the retrieval algorithm to study its limitations and sensitivities, and it would be a rather weakly defended conclusion in the context of this manuscript.

**Comment 2.3.16:** *Fig 2, explain what unc, sic etc. mean within figure caption.*

**Response:** Done. We have also added these explanations to the main text.

**Comment 2.3.17:** *Fig 2, is it impossible to get forecast SIT below 0.3 m when SIC is low (i.e. when SMOS-SIT is around 0)? Why?*

**Response:** No it is not impossible, see Figure 1 in this response. The apparent gap is due to the filtering applied, where only data points with SIC > 30% are used.

**Comment 2.3.18:** *Fig 3, does (c) show saturation in the SMOS-SIT signal above approximately 0.5 m? Plateaus above this value, so no sensitivity from L-band signal?*

**Response:** It should not be lack of sensitivity, because all data points shown have a SMOS-SIT saturation ratio of below 90% (i.e. the retrieved SIT is 90% of the maximally retrievable SIT under these conditions). However, there could be a conceptual problem with the saturation ratio provided with the SMOS-SIT product.

**Comment 2.3.19:** *Fig 3, doing this for only one years winter enhances the possibility for anomalous ice conditions to explain the departures between observed and predicted IT. What do these look like for multiple years? Your arguments would be more convincing if similar patterns of regional biases were found in several/all years.*

**Response:** We fully agree and have taken this excellent suggestion on board. We have updated Figure 3 to include data from *all* winters, and find that the departure characteristics appear in all years.

**Comment 2.3.20:** *Fig 4, Mark on a map either here or on Fig 1. Adding a panel of SIC would be very useful for analysis.*

**Response:** We have marked the locations in Figure 1 as suggested. The SIC time series is already shown in Figure 5b, we have added a reference to the caption of Figure 4.

**Comment 2.3.21:** *Fig 4, "added and subtracted" what? Uncertainty?*

**Response:** We have added the word "uncertainty" to the caption. Apologies for the omission.

**Comment 2.3.22:** *Fig 5c, why does snow depth appear to drop considerably throughout the season?*

**Response:** It only drops in SMOS-SIT, not in ORAS5. The simple reason for the snow thickness drop in SMOS-SIT is that the retrieval algorithm assumes a snow thickness that is a piecewise linear function of ice thickness (Tian-Kunze et al., 2014). Thus, snow thickness in SMOS-SIT is not an independent parameter. In this case, one might argue that this leads to an unrealistic snow thickness. However, sensitivity of retrieved SIT to snow thickness is relatively small.

**Comment 2.3.23:** *Fig 5e, ice emissivity masked by overlying snow?*

**Response:** Dry snow is transparent in L-band and therefore does not mask the ice emissivity. Snow only enters the SMOS-SIT retrieval algorithm through its thermal insulation qualities: more snow means the ice is better insulated against the cold atmosphere, and bulk ice temperature tends to be higher, which changes the ice emissivity.

**Comment 2.3.24:** *Fig 6, remove "none".*

**Response:** Fixed.

**References**

Ivanova, N., Pedersen, L. T., Tonboe, R. T., Kern, S., Heygster, G., Lavergne, T., Sørensen, A., Saldo, R., Dybkjær, G., Brucker, L., and Shokr, M.: Inter-comparison and evaluation of sea ice algorithms: towards further identification of challenges and optimal approach using passive microwave observations, The Cryosphere, 9, 1797–1817, doi:10.5194/tc-9-1797-2015, URL `http://www.the-cryosphere.net/9/1797/2015/tc-9-1797-2015.html`, 2015.

Landy, J. C., Ehn, J. K., Babb, D. G., Theriault, N., and Barber, D. G.: Sea ice thickness in the Eastern Canadian Arctic: Hudson Bay Complex & Baffin Bay, Remote Sensing of Environment, 200, 281–294, doi:10.1016/J.RSE.2017.08.019, URL `http://www.sciencedirect.com/science/article/pii/S0034425717303887`, 2017.

Maaß, N.: Remote sensing of sea ice thickness using SMOS data, Ph.D. thesis, University of Hamburg, 2013.

Richter, F., Drusch, M., Kaleschke, L., Maaß, N., Tian-Kunze, X., and Mecklenburg, S.: Arctic sea ice signatures: L-Band brightness temperature sensitivity comparison using two radiation transfer models, The Cryosphere Discussions, 2016, 1–22, doi:10. 5194/tc-2016-273, URL `http://www.the-cryosphere-discuss.net/tc-2016-273/`, 2016.

Tian-Kunze, X., Kaleschke, L., Maaß, N., Mäkynen, M., Serra, N., Drusch, M., and Krumpen, T.: SMOS-derived thin sea ice thickness: algorithm baseline, product specifications and initial verification, The Cryosphere, 8, 997–1018, doi: 10.5194/tc-8-997-2014, URL `http://www.the-cryosphere.net/8/997/2014/http://www.the-cryosphere.net/8/997/2014/tc-8-997-2014.html`, 2014.

Uotila, P., Goosse, H., Haines, K., Chevallier, M., Barthelemy, A., Bricaud, C., Carton, J., Fuckar, N., Garric, G., Iovino, D., Kauker, F., Korhonen, M., Lien, V. S., Marnela, M., Massonnet, F., Mignac, D., Peterson, K. A., Sadikni, R., Shi, L., Tietsche, S., Toyoda, T., Xie, J., and Zhang, Z.: An assessment of ten ocean reanalyses in the polar regions, Climate Dynamics, (under rev, 1–64, 2018.

[Figure]

Figure 1: Joint frequency distribution of (a) ORAS5 SIC and SIT and (b) SMOS-SIT and ORAS5 SIT calculated for 15 November 2016 (the date for which the upper row of maps in Figure 1 of the manuscript is shown). All data points with a valid SMOS-SIT value have been considered, no filter was applied.

[Figure]

Figure 2: Time series of ice thickness in SMOS-SIT (blue line) and ORAS5 (red line) for the winters 2011/12 to 2016/17. Thickness is calculated from all data points within the box 80W–64W, 67N–75N, which corresponds to the Wester Baffin Bay area as defined in Landy et al. (2017).

[Figure]

Figure 3: Normalized joint frequency distribution (scatter density) of pairs of SMOS-SIT retrieval uncertainty and SMOS-SIT–ORAS5 departures; (a) October to December 2011–2017, (b) February to April 2012–2017. All data points with a valid SMOS-SIT value have been considered, no filter was applied.

[Figure]

Figure 4: Normalized joint frequency distribution (scatter density) of pairs of OSTIA SIC and SMOS-SIT SIT within the box 80W–64W, 67N–75N (roughly corresponding to the Western Baffin Bay as defined by Landy et al. (2017)); (a) October to December 2011–2017, (b) February to April 2012–2017. All data points with a valid SMOS-SIT value have been considered, no filter was applied.

---

## Author Response (AR2)

**Authors response to editor's comments for manuscript tc-2017-247**

*We would like to thank the editor for reading the manuscript and suggesting some final changes, which we have all implemented as described below. The numbering of pages and lines in the editor's comments and our response refers to the version of the manuscript with highlighted changes.*

**Comment 1:** *Specify up front what you mean by thin ice (including in the abstract)*

**Response:** Thank you very much for spotting this issue. The term "thin" is indeed qualitative and might be ambiguous. In this manuscript, we implicitly equate "thin" with "possibly retrievable by L-band radiometry". Although the maximum retrievable ice thickness for L-band varies a lot depending on the surface conditions, we think that an upper limit of 1 m is a reasonable threshold. We have made this explicit in the abstract, and in several appropriate locations throughout the manuscript (P3L8, P16L3, P19L1).

**Comment 2:** *Page 3, first paragraph feels disjointed from the second paragraph. You mention L-band in the first paragraph but go into more detail in the second paragraph, but it's strange to say for the first time SMOS has allowed for L-band measurements. I think some better flow can be done between these two paragraphs.*

**Response:** We have revised the end of the first paragraph and the beginning of the second paragraph on page 3 to accommodate this comment.

**Comment 3:** *Page 8, line 26, typo. Should be note. Also, the thickness means the mean grid cell thickness even if part is open water, or the thickness of the ice that is actually there? It's always good to specify since PIOMAS for example includes open water fraction in the mean grid cell thickness, whereas CPOM CS2 data does not. I think it's important to also point out how this compares with how climate models define it so that users of SMOS can correctly compare with thin ice in GCMs for example.*

**Response:** Typo fixed. Thank you for pointing out that the term "grid-cell mean ice thickness" is ambiguous, we concur. We mean it in the sense of *sea ice volume per area* (variable *sivol* in CMIP6 definitions). We have clarified this on page 8, line 26ff., and have added a reference to Notz et al. (2016). We have revised the manuscript in several locations to avoid the use of the term "grid-cell mean ice thickness".

**Comment 4:** *I would always suspect SMOS thickness to be lower than ORAS5 near the ice edge because of open water fraction. But wouldn't the ORAS5 also capture the open water? Have both the SMOS and ORAS5 been gridded to the same grid? Also I don't fully understand why the SMOS data is on a 12.5 km grid because the actual spatial resolution is coarser than this.*

**Response:** Regarding the open water fraction, we agree that this is a major outstanding issue that causes SMOS-SIT to be biased towards lower thicknesses. This has been explicitly acknowledged in Tian-Kunze et al. (2014) and discussed in this manuscript (e.g. page 11, lines 17–25). As detailed in Section 2.2 of the manuscript, ORAS5 models and assimilates sea-ice concentration, so it does capture open water.

Regarding the gridding, we have interpolated the ORAS5 data to the SMOS-SIT grid for comparison. The same interpolation would be applied to compute observation–model departures in a data assimilation framework. SMOS-SIT is provided on such a fine grid for the following reason: The SMOS L1C data are given on the Discrete Global Grid (DGG) system. The DGGs are fixed Earth grid coordinates of the ISEA hexagonal grid centers which have a spatial distance of 15 km. The use of a 12.5 km grid is a reasonable choice to comply both with the commonly used polar stereographic (or EASE) grid and Nyquist's sampling theorem which requires a certain amount of oversampling to avoid aliasing. We have added a comment on the choice of the grid for SMOS-SIT on page 5 line 24.

**References**

[revised manuscript text omitted]